# Certifiably Robust Model Evaluation in Federated Learning under Meta-Distributional Shifts

**Amir Najafi** [1]  **Samin Mahdizadeh Sani** [2]  **Farzan Farnia** [3]

## Abstract

We address the challenge of certifying the performance of a federated learning model on an unseen target network using only measurements from the source network that trained the model. Specifically, consider a source network "A" with $K$ clients, each holding private, non-IID datasets drawn from heterogeneous distributions, modeled as samples from a broader meta-distribution $\mu$. Our goal is to provide certified guarantees for the model's performance on a different, unseen network "B", governed by an unknown meta-distribution $\mu'$, assuming the deviation between $\mu$ and $\mu'$ is bounded—either in *Wasserstein* distance or an $f$-*divergence*. We derive worst-case uniform guarantees for both the model's average loss and its risk CDF, the latter corresponding to a novel, adversarially robust version of the Dvoretzky–Kiefer–Wolfowitz (DKW) inequality. In addition, we show how the vanilla DKW bound enables principled certification of the model's true performance on unseen clients within the same (source) network. Our bounds are efficiently computable, asymptotically minimax optimal, and preserve clients' privacy. We also establish non-asymptotic generalization bounds that converge to zero as $K$ grows and the minimum per-client sample size exceeds $\mathcal{O}(\log K)$. Empirical evaluations confirm the practical utility of our bounds across real-world tasks. The project code is available at: github.com/samin-mehdizadeh/Robust-Evaluation-DKW

[1]Department of Computer Engineering, Sharif University of Technology, Tehran, Iran (Corresponding author) [2]Department of Electrical and Computer Engineering, University of Tehran, Tehran, Iran [3]Department of Computer Science and Engineering, The Chinese University of Hong Kong (CUHK), Hong Kong. Correspondence to: Amir Najafi <amir.najafi@sharif.edu>, Samin Mahdizadeh Sani <samin.mahdizadeh@ut.ac.ir>, Farzan Farnia <farnia@cse.cuhk.edu.hk>.

*Proceedings of the 42$^{nd}$ International Conference on Machine Learning*, Vancouver, Canada. PMLR 267, 2025. Copyright 2025 by the author(s).

## 1. Introduction

The distributed nature of modern learning environments, where local datasets are scattered across clients in a network, presents significant challenges for the machine learning community. Federated learning (FL) addresses some of these challenges by enabling clients to collaboratively train a decentralized model through communications with a central server (McMahan et al., 2017; Liu et al., 2024). A major obstacle in FL is the heterogeneity of data distributions among clients. This non-IID nature of client data not only impacts the training stage but also complicates the evaluation of trained models, especially when applied to unseen clients from the same or different networks (Ye et al., 2023; Zawad et al., 2021). In this paper, we focus on the latter problem.

Consider a network with $K$ clients, each possessing their own private dataset. For $k \in [K]$, the $k$th client's dataset is assumed to include $n_k \geq 1$ samples from a unique and unknown distribution $P_k$, which forms an empirical and private estimate $\widehat{P}_k$. Note that the distributions $P_1, \ldots, P_K$ may be highly heterogeneous and distant from one another. In this setting, assume the server wants to assess the performance of a given machine learning model $h$ on this network. Let the *risk* function $R(h, \widehat{P}_k)$ denote the loss of $h$ when evaluated over the private data samples of the $k$th client, i.e., $\widehat{P}_k$. This quantity can be queried by the server by sending $h$ to client $k$ and requesting its local loss value, without directly accessing any of the private data samples in client $k$'s dataset. Hence, the server can collect all $K$ loss values and empirically approximate the average performance of $h$ over the network, using various metrics. One common approach is the average loss: $\frac{1}{K} \sum_{k=1}^{K} R(h, \widehat{P}_k)$, which measures the empirical average performance of the model in the network.

Another approach is the loss CDF, representing the percentage of clients whose loss exceeds a given threshold $\lambda$: $\frac{1}{K} \sum_{k=1}^{K} \mathbb{1}\big(R(h, \widehat{P}_k) \geq \lambda\big)$, which is useful for estimating the quantiles of the risk (see (Laguel et al., 2021)). For a recent related work on the use of super-quantiles in FL with heterogeneous clients, see (Pillutla et al., 2024). While that study focuses on optimization and convergence aspects, our work provides a theoretical treatment of generalization.

A critical challenge arises when certifying the performance of $h$ on a different and unseen network with completely different clients. Mathematically, assume a large group of $T \gg 1$ new and unseen clients with corresponding heterogeneous distributions $Q_1, \ldots, Q_T$, which may differ substantially from one another and also from $P_k$'s. This corresponds to a *beta testing* scenario, where, for instance, a product owner may test a product on a small subset of a network before wider deployment (Reisizadeh et al., 2020; Ma et al., 2024). Our goal is to certify the same performance metrics over $Q_i$s without accessing them.

Clearly, without a connection between $P_k$'s and $Q_i$'s, the certification task is fundamentally infeasible. To address this, we can rely on the following common assumption in many beta testing scenarios: $P_{1:K}$ and $Q_{1:T}$ are all independent samples from the same *meta-distribution* $\mu$ (a distribution over distributions), which governs higher-level factors influencing the clients, such as cultural or geographical attributes (Yuan et al., 2021; Ajay et al., 2022; Wu et al., 2024a; Chen et al., 2023; Patel et al., 2022). In this setting, the empirical average risk and empirical risk CDF over $P_k$'s provide unbiased estimates for those of the unseen clients, i.e., $Q_i$'s. Still, a natural question would be how large $K$ or either of $n_k$'s need to be such that the mentioned estimates become statistically reliable?

In this work, we propose a novel approach by leveraging the Dvoretzky–Kiefer–Wolfowitz (DKW) inequality (Dvoretzky et al., 1956) that uniformly bounds the CDF estimation error from empirical observed samples. Note that in our setting, the CDF is considered for the performance metric of clients drawn from meta distribution $\mu$. Therefore, the observed clients' performance values lead to a proxy *empirical CDF*, and we want to provide an upper-bound on the *true CDF* to provide a performance guarantee for an unseen client $Q_i \sim \mu$. Our approach is to extend the DKW inequality to bound the gap between the empirical and population (true) CDFs. Figure 2 illustrates our DKW inequality-based approach and how it can provide a certified upper-bound on the true CDF of unseen clients' performance.

On the other hand, in real-world applications, target clients can belong to distinct populations with considerable cultural or behavioral differences compared to those observed in the evaluation phase. For example, an app developer might be limited to test a mobile application on users from Hong Kong, but ultimately hopes to release it for users from New York. This scenario corresponds to a meta-distributionally robust evaluation, where the objective is to evaluate the performance of $h$ under worst-case shifts from $\mu$ to $\mu'$. Here, $\mu$ and $\mu'$ represent unknown meta-distributions, with $K$ empirical samples from $\mu$ corresponding to $P_1, \ldots, P_K$. Unfortunately, $\mu'$ is completely unseen, however, it is assumed to deviate at most $\varepsilon$ from $\mu$, for some known $\varepsilon > 0$.

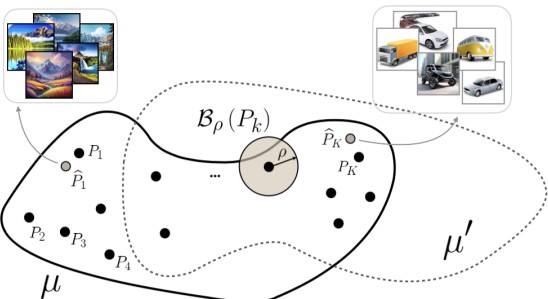

*Figure 1.* A graphical illustration of meta-distributional shifts between two client networks governed by $\mu$ and $\mu'$. Meta-shifts shifts may only denote changes in densities ($f$-divergence) or can involve support as well (Wasserstein shifts). Each client $P_k$ is a sample from $\mu$, with $\widehat{P}_k$s as its empirical counterpart based on a local datasets. The client distributions $P_k$ can vary significantly, such as one having primarily landscape images while another contains mostly vehicle images.

Again, without this assumption certifying the performance of $h$ under $\mu'$ is mathematically impossible. The degree of deviation ($\varepsilon$) between $\mu$ and $\mu'$ can be quantified using metrics such as Kullback-Leibler (KL) divergence or Wasserstein distance, both interpreted in the context of meta-distributions. KL, or more generally, an $f$-divergence captures situations where client types in one meta-distribution are reweighted in another (Mehta et al., 2024), whereas Wasserstein distance accounts for the emergence of entirely new client types, representing novel regions of the meta-distribution's support (Wang et al., 2022; Kuhn et al., 2019). For example, smartphone users in a coastal area may often take pictures of the sea, whereas residents in mountainous regions cannot do so unless they travel. See Figure 1 for more details. We are not aware of any prior theoretical bounds for meta-distributionally robust evaluation (or beta testing) in federated settings. In any case, a review of prior works on similar scenarios of non-IID federated learning and model evaluation is provided in Section A.

**Our Contribution:** Our problem, formally defined in Section 3, is to address the robust evaluation of any given $h$ under the above-mentioned scenario. Section 4 presents some initial results for the non-robust evaluation when $\mu = \mu'$, i.e., standard beta testing. Theorem 4.1 proves non-asymptotic concentration bounds for both the empirical average and CDF of the risk, with exponential convergence as both $K$ and $\min_k n_k$ increase. Notably, our convergence results for the loss CDF are *uniform* over thresholds $\lambda$, meaning they hold for all $\lambda \in \mathbb{R}$ simultaneously, via utilizing classical tools in statistics, including the DKW inequality and Glivenko-Cantelli theorem (Dvoretzky et al., 1956).

Next, we provide robust bounds on the average loss and/or risk CDF when target network is adversarially shifted from the source. We extend the concept of distributional robust-

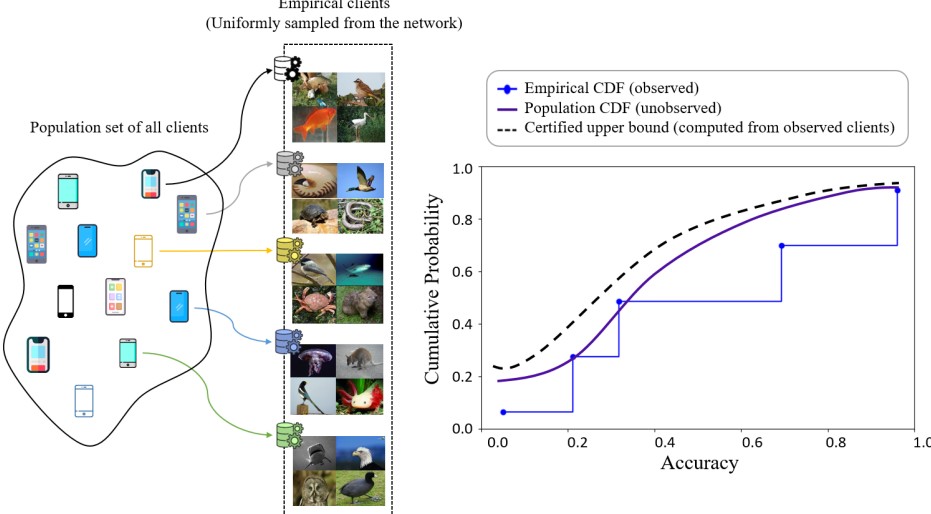

*Figure 2.* Our theoretical bounds are estimated from a set of seen users in a federated network governed by meta-distribution $\mu$. We extend the application of DKW inequality to provide a certified upper-bound on the true CDF of the performance score of unseen clients following the same meta-distribution $\mu$.

ness to meta-distributional robustness, considering both $f$-divergences and Wasserstein-type adversaries, respectively in Sections 5 and 6, to provide worst-case guarantees.

The majority of existing robust evaluation schemes require aggregating all private datasets in a central server for a collective adversarial manipulation of all the samples (Liu et al., 2024; Reisizadeh et al., 2020; Sinha et al., 2018), which severely violates user privacy. Another category of approaches simply compute the average ordinary risk of $h$ in the source network, but are then forced to add a non-vanishing ($\varepsilon$-dependent) generalization gap that does not go to zero even when $K, n_{1:K}$ grow to infinity (see Section A and also (Rahimian and Mehrotra, 2019; Zeng and Lam, 2022)). The latter approaches fail to take into account the possible inherent wellness or robustness of $h$, thus usually result into excessively inflated risk certificates.

We prove that *globally* robust evaluations under both $f$-divergence and Wasserstein regimes can be done in a federated manner as long as: i) server can query the *local* ordinary or adversarial (only for Wasserstein attacks) loss of $h$ from each client, and ii) server can repeat such queries with potentially various attack budgets for polynomially many times. Note that we do not need to directly access client's data. Our evaluation guarantees are asymptotically minimax optimal, and have vanishing generalization gaps which converge to zero as both $K$ and $(\min_k\ n_k)/\log K$ grow. Our robust and uniform bounds on risk CDF in Section 5 are based on a novel extension of DKW inequality to adversarial scenarios, which is extensively detailed in Section B. The numerical results on standard datasets in Section 8 validate the tightness of our bounds.

## 2. Notations and Preliminaries

For $K \in \mathbb{N}$, $[K]$ denotes the set $\{1, 2, \ldots, K\}$. Consider two measurable spaces $\mathcal{X}$ and $\mathcal{Y}$, referred to as the *feature* and *label* spaces, respectively. Typically, we assume $\mathcal{X} \subseteq \mathbb{R}^d$ for some $d \in \mathbb{N}$, while $\mathcal{Y}$ may be $\{\pm 1\}$ for binary classification tasks or $\mathbb{R}$ in regression problems. However, we make no extra assumptions regarding $\mathcal{X}$ and $\mathcal{Y}$. We define the joint feature-label space as $\mathcal{Z} \triangleq \mathcal{X} \times \mathcal{Y}$. Let $\mathcal{M}(\mathcal{Z})$ denote the set of all probability measures supported on $\mathcal{Z}$. Each $P \in \mathcal{M}(\mathcal{Z})$ corresponds to a joint measure over a random feature vector $\boldsymbol{X} \in \mathcal{X}$ and its associated random label $y \in \mathcal{Y}$. The expectation operator with respect to a measure $P$ is denoted by $\mathbb{E}_P$. We refer to $\mu$ as a meta-distribution over $\mathcal{Z}$, expressed as $\mu \sim \mathcal{M}^2(\mathcal{Z}) \triangleq \mathcal{M}(\mathcal{M}(\mathcal{Z}))$, where each sample from $\mu$ is itself a probability measure over $\mathcal{Z}$. In other words, a meta-distribution is a distribution over distributions supported on $\mathcal{Z}$. In our work, $\mu$ models the heterogeneity and non-IID-ness of the clients, since independent samples from $\mu$ represent different data distributions shifted from one another.

For simplicity, we abbreviate $\mathcal{M}(\mathcal{Z})$ as $\mathcal{M}$. For any two measures $P, Q \in \mathcal{M}$, let $\mathcal{D}(P, Q) \in \mathbb{R}_{\geq 0}$ denote any given distance or divergence between the two distributions such as KL or Wasserstein distance. In this context, let $\mathcal{B}_\rho$ denote an $\rho$-distributional ambiguity ball for $\rho \geq 0$. Mathematically, for any measure $P \in \mathcal{M}$:

$$\mathcal{B}_\rho(P) \triangleq \{Q \in \mathcal{M} \mid \mathcal{D}(P, Q) \leq \rho\}, \qquad (1)$$

which represents the set of distributions within $\rho$ distance/divergence from $P$ according to $\mathcal{D}$. Similarly, meta-distributional ambiguity balls $\mathcal{G}_\varepsilon(\mu)$ are defined as the set of meta-distributions over $\mathcal{M}^2$ within an $\varepsilon$ distance

from a base meta-distribution $\mu$. Mathematically, we have $\mathcal{G}_\varepsilon(\mu) \triangleq \left\{ \mu' \in \mathcal{M}^2 \mid \widetilde{\mathcal{D}}(\mu, \mu') \leq \varepsilon \right\}$. Here, the *deviation* $\widetilde{\mathcal{D}}$ can be any properly defined $f$-divergence or a Wasserstein metric between the meta-distributions in $\mathcal{M}^2$. In particular, the transportation cost in the Wasserstein distance can itself be a Wasserstein metric on $\mathcal{M}$ in its ordinary sense.

Let $h : \mathcal{X} \to \mathcal{Y}$ represent a hypothesis (e.g., a classifier) that maps the feature space $\mathcal{X}$ onto the label space $\mathcal{Y}$ [1]. Additionally, assume a fixed loss function $\ell(y, \widehat{y})$, which assigns a loss value to each pair of actual and predicted labels, $y$ and $\widehat{y}$, respectively. The expected loss, or *Risk*, of a hypothesis $h$ w.r.t. a data distribution $P \in \mathcal{M}$ is defined as $R(h, P) \triangleq \mathbb{E}_P \{\ell(y, h(\boldsymbol{X}))\}$. Here, $P$ could be the true distribution or an empirical approximation obtained from a dataset, usually denoted by $\widehat{P}$. The *adversarial* risk of $h$ w.r.t. a base measure $P$ (or empirical $\widehat{P}$) and a robustness radius $\varepsilon \geq 0$, is formulated as (Sinha et al., 2018):

$$R_{\varepsilon-\mathrm{adv}}(h, P) \triangleq \sup_{Q \in \mathcal{B}_\varepsilon(P)} \mathbb{E}_Q[\ell(y, h(\boldsymbol{X}))], \qquad (2)$$

and denotes the worst risk over all distributions in a $\varepsilon$-neighborhood of $P$ according to $\mathcal{D}$. Similarly, the meta-distributionally robust loss of $h$ with respect to a base meta-distribution $\mu$ is defined as:

$$\sup_{\mu' \in \mathcal{G}_\varepsilon(\mu)} \mathbb{E}_{P \sim \mu'} [\mathbb{E}_P [\ell(y, h(\boldsymbol{X}))]]. \qquad (3)$$

The geometry of the ball $\mathcal{G}_\varepsilon$ in $\mathcal{M}^2$ can be determined using various application-specific divergences or metrics over $\mathcal{M}^2$. In Section 5, we deal with $f$-divergence balls, while Wasserstein meta-distributional balls (using the ordinary Wasserstein metric as their transportation cost) are utilized in Section 6. We also discuss how to practically implement such loss values in Section 7 and Appendix F.

## 3. Problem Definition

This section formally defines our problem. First, we outline the data generation process, privacy constraints, and specify the query policy that governs communication between clients and the server.

**Data Generation:** Consider $K \in \mathbb{N}$ clients connected to a central server, where client $k \in [K]$ has a unique data distribution $P_k \in \mathcal{M}(\mathcal{Z})$. We assume $P_1, P_2, \ldots, P_K$ are heterogeneous samples of an unknown meta-distribution $\mu$ over $\mathcal{Z}$. No one knows $P_k$s, however, client $k$ has access to a dataset $D_k$ of size $n_k \geq 1$, which contains independent samples from $P_k$, i.e.,

$$D_k \triangleq \left\{ \left( \boldsymbol{X}_i^{(k)}, y_i^{(k)} \right) \mid i \in [n_k] \right\} \sim P_k^{\otimes n_k}. \qquad (4)$$

---

[1]It should be noted that our work goes beyond this limitation and can be applied to any supervised or unsupervised machine learning task.

Let $\widehat{P}_k$ denote the empirical version of $P_k$ based on the private samples in $D_k$, which are known only to client $k$.

**Server-Client Query Policy:** The server can query each of the $K$ clients by sending a model $h$ and a robustness radius $\rho \geq 0$ to client $k \in [K]$. In response, the client returns the adversarial loss of $h$ around $\widehat{P}_k$:

$$\widehat{\mathsf{QV}}_k(h, \rho) \triangleq \sup_{Q \in \mathcal{B}_\rho(\widehat{P}_k)} \mathbb{E}_Q[\ell(y, h(\boldsymbol{X}))]. \qquad (5)$$

The type of distributional ball $\mathcal{B}_\rho(\cdot)$ can be defined using any user-defined divergence or metric over $\mathcal{M}$. When the robustness radius is unspecified, the client assumes it is zero, and returns the non-robust loss, i.e., $\widehat{\mathsf{QV}}_k(h) = \widehat{\mathsf{QV}}_k(h, 0)$. We later show that the local adversarial loss (when $\rho > 0$) is not needed for $f$-divergence-based guarantees and only appears in Section 6 which focuses on Wasserstein shifts. Each client $k$ accepts a maximum number of queries, referred to as the *query budget*.

**Our Problem Setup:** Assume the server sends a model $h$ to the clients and requests a number of robust or non-robust loss values for several arbitrary robustness radii. Server's ultimate goal is to use these values to provide a meta-distributionally robust upper bound for the average or CDF of the loss of $h$. Mathematically speaking, the objective is to *efficiently* compute empirical values $\widehat{E}_1(\varepsilon)$ and $\widehat{E}_2(\varepsilon, \lambda)$ such that the following bounds hold with high probability over the sampling of clients and datasets:

$$\sup_{\mu' \in \mathcal{G}_\varepsilon(\mu)} \mathbb{E}_{P \sim \mu'} [\mathbb{E}_P [\ell(y, h(\boldsymbol{X}))]] \leq \widehat{E}_1(\varepsilon) + \zeta, \qquad (6)$$

$$\sup_{\mu' \in \mathcal{G}_\varepsilon(\mu)} \mu' \Big( \mathbb{E}_P [\ell(y, h(\boldsymbol{X}))] \geq \lambda \Big) \leq \widehat{E}_2(\varepsilon, \lambda) + \zeta'.$$

The generalization gaps $\zeta, \zeta'$ should vanish as both $K$ and $\min_k n_k$ increase asymptotically. Also, the second high-probability bound needs to hold for all threshold values $\lambda$, simultaneously (or *uniformly*). We consider both $f$-divergence and Wasserstein metrics to specify the geometry of the meta-distributional ambiguity ball $\mathcal{G}_\varepsilon(\mu)$.

From an algorithmic perspective, $\widehat{E}_{1:2}$ must be computable using only the private server-client query policy $\widehat{\mathsf{QV}}_k(h, \rho)$ for various robustness radii $\rho$. Server decides the number of queries for each client, as well as the value of $\rho$ for each query. However, the computational cost of evaluating each query at the client side, and the total number of queries per client should increase at most polynomially with parameters.

## 4. Non-Robust (Ordinary) Guarantees

In this section, we solve the problem setting of section 3 in the non-robust regime, i.e., when $\mu' = \mu$ or equivalently $\varepsilon = 0$. This scenario corresponds to the standard

beta testing, which has been extensively applied in practice (Chen et al., 2024). Simple statistical bounds can establish the high-probability concentration of the empirical average $\frac{1}{K} \sum_{k=1}^{K} R(h, \widehat{P}_k)$ around its true mean w.r.t. $\mu$, as long as the loss function $\ell(\cdot)$ is measurable and bounded (e.g., let $\ell$ be 1-bounded such as the 0-1 loss). However, the same argument does not directly extend to the loss CDF, as the bounds must hold uniformly for all threshold values $\lambda \geq 0$. Without uniformity, such guarantees are significantly weakened. By "uniform," we mean that, with high probability over the sampling of the $K$ users, the worst-case deviation between the empirical and true tail probabilities is consistently bounded. This challenge is resolved using a classic result on the convergence of empirical CDFs, specifically the Dvoretzky–Kiefer–Wolfowitz (DKW) inequality and the Glivenko–Cantelli theorem (see Section B, and in particular Lemma B.1). The following theorem summarizes our main findings in the non-robust setting.

**Theorem 4.1.** *Let $\mu$ be a meta-distribution, and assume $P_1, \ldots, P_K$ be $K$ independent instances of $\mu$, while $\widehat{P}_k$ for $k \in [K]$ represent their empirical counterparts, formed using $n_k$ independent private samples from the $k$-th client. Let $h : \mathcal{X} \to \mathcal{Y}$ be any model and $\ell$ be any measurable and 1-bounded loss function. Then, for any $\delta > 0$, the following holds with probability at least $1 - \delta$:*

$$\mathbb{E}_\mu \left[ \mathbb{E}_P \left( \ell \left( y, h \left( \boldsymbol{X} \right) \right) \right) \right] \leq \frac{1}{K} \sum_{k=1}^{K} \widehat{\mathsf{QV}}_k (h) + \qquad (7)$$

$$\sqrt{\log \left( K \delta^{-1} \right)} \mathcal{O} \left( \frac{1}{\sqrt{K}} + \frac{1}{K} \sum_{k=1}^{K} n_k^{-1/2} \right).$$

*For the loss CDF, the following bound holds with probability at least $1 - \delta$ uniformly for all $\lambda \in \mathbb{R}$:*

$$\mu \left( \mathbb{E}_P \left[ \ell \left( y, h \left( \boldsymbol{X} \right) \right) \right] \geq \lambda \right) \leq \qquad (8)$$

$$\frac{1}{K} \sum_{k=1}^{K} \mathbb{1} \left( \widehat{\mathsf{QV}}_k(h) \geq \lambda - \sqrt{\frac{\log \frac{(K+1)}{\delta}}{2 n_k}} \right) + \sqrt{\frac{\log \frac{2(K+1)}{\delta}}{2K}}.$$

The proof is given in Appendix C. By ignoring poly-logarithmic terms, the empirical means over both the $K$ users and the $n_k$ samples from each client $k$ converge at a rate of $(\min \{K, \min_k \ n_k / \log K\})^{-1/2}$. In both inequalities, the left-hand sides represent strong statistical quantities, while the right-hand sides consist of: i) vanishing generalization gaps, plus ii) empirical values that can be fully evaluated based on users' private datasets and using the private query policy described earlier (i.e., the $\widehat{\mathsf{Q}}_i(h)$ values). Additionally, each client needs to be queried only once.

In Section 7, we discuss the computational complexity, certificates of privacy, and tightness of the all the bounds in our work. Before that, Sections 5 and 6 extend Theorem 4.1 to the robust setting, where $\mu'$ can be adversarially perturbed from $\mu$ by either an $f$-divergence or a Wasserstein adversary.

## 5. $f$-Divergence Meta-Distributional Shifts

In this section, we provably bound the meta-distributionally robust performance of $h$, in terms of the average risk and risk CDF.

**Definition 5.1.** Consider two meta-distributions $\mu, \mu' \in \mathcal{M}^2$ where $\mu'$ is absolutely continuous with respect to $\mu$. Let $f : [0, \infty) \to [-\infty, \infty]$ be a convex function such that $f(x)$ is finite for all $x > 0$, $f(1) = 0$, and $f(0) = \lim_{t \to 0^+} f(t)$ (which could be infinite). The $f$-divergence between $\mu$ and $\mu'$ is defined as:

$$\mathcal{D}_f(\mu' \| \mu) = \int_{P \in \mathcal{M}} f \left( \frac{\mathrm{d}\mu'(P)}{\mathrm{d}\mu(P)} \right) \mathrm{d}\mu(P). \qquad (9)$$

Also, for $\varepsilon \geq 0$, the $f$-divergence ball $\mathcal{G}_\varepsilon^{f-\mathrm{div}}(\mu)$ is defined as $\{\mu' \mid \mathcal{D}_f(\mu' \| \mu) \leq \varepsilon\}$, which describes a neighborhood around $\mu$ where the divergence does not exceed $\varepsilon$.

Our goal is to provide a theoretical guarantee for the loss of $h$ under the worst-case meta-distribution $\mu' \in \mathcal{G}_\varepsilon^{f-\mathrm{div}}(\mu)$, using only the query values from client samples in the source network governed by $\mu$. Since clients generated by $\mu'$ may follow different densities compared to $\mu$, it is natural to reweight the robust loss to achieve a robust upper bound. The following theorem formalizes this idea by reweighting the query values using coefficients $\alpha_k$, $k \in [K]$, and optimizing for the worst-case weights, provided they remain close to uniform weights. This approach yields a minimax-optimal bound with a vanishing generalization gap.

**Theorem 5.2.** *Assume an unknown meta-distributions $\mu \in \mathcal{M}^2$, let $\varepsilon \geq 0$, and consider $f(\cdot)$ to be as in Definition 5.1. Let $h : \mathcal{X} \to \mathcal{Y}$ be any model and $\ell$ be any measurable and 1-bounded loss function. Assume $P_1, \ldots, P_K$ represent $K$ independent and unknown sample distributions from $\mu$. Accordingly, let $\widehat{P}_k$ for $k \in [K]$ represent their empirical counterparts, formed using $n_k$ independent private samples from the $k$-th client. Let $\Lambda \triangleq 1 + \kappa \varepsilon$. Define $\widehat{B}^*$ as:*

$$\widehat{B}^*(\varepsilon) \triangleq \sup_{0 \leq \alpha_1, \ldots, \alpha_K \leq \Lambda} \frac{1}{K} \sum_{k=1}^{K} \alpha_k \widehat{\mathsf{QV}}_k (h) \qquad (10)$$

$$\text{subject to} \quad \left| \frac{1}{K} \sum_{k=1}^{K} \alpha_k - 1 \right| \leq c_1 \sqrt{\log \left( \delta^{-1} \right) / K},$$

$$\frac{1}{K} \sum_{k=1}^{K} f(\alpha_k) \leq \varepsilon + c_2 \sqrt{\log \left( \delta^{-1} \right) / K},$$

*where constants $\kappa, c_1, c_2 \geq 0$ are known and depend on $f(\cdot)$. Then, for any $\delta > 0$, the following bound holds with*

*probability at least* $1 - \delta$*:*

$$\sup_{\mu' \in \mathcal{G}_\varepsilon^{f-\mathrm{div}}(\mu)} \mathbb{E}_{P \sim \mu'} \left[ \mathbb{E}_P \left[ \ell \left( y, h \left( \boldsymbol{X} \right) \right) \right] \right] \leq \widehat{B}^*(\varepsilon) \quad (11)$$

$$+ \sqrt{\log \left( K \delta^{-1} \right)} \mathcal{O} \left[ \frac{1}{\sqrt{K}} + \frac{1}{K} \sum_{k=1}^{K} n_k^{-1/2} \right].$$

The proof including the formulations for $\kappa, c_1, c_2$ are provided in Appendix D. This theorem establishes a robust bound on the expected loss under meta-shifts using $\widehat{B}^*(\varepsilon)$, along with a vanishing generalization gap with a decay rate of $\tilde{\mathcal{O}} \left( \min\{K, n_{1:K}\}^{-1/2} \right)$. The empirical quantity $\widehat{B}^*(\varepsilon)$ is derived from a convex optimization problem performed server-side. In Section 7, we discuss all of aspects of computing this quantity. Also, it worth noting that as both $\varepsilon, \frac{1}{K}$ tend to zero, the bound becomes increasingly similar to the non-robust case of Theorem 4.1, which should be expected. As discussed before, a key advantage of our result is that the generalization gap lacks any *non-vanishing* $\varepsilon$-dependent term. Next, we present our main result for robust CDF estimation of the loss under KL meta-shifts:

**Theorem 5.3.** *Assume the setting of Theorem 5.2. For any* $\lambda \in \mathbb{R}$*, let us define the empirical value* $\widehat{J}^* (\varepsilon, \lambda)$ *as*

$$\sup_{0 \leq \alpha_{1:K} \leq \Lambda} \frac{1}{K} \sum_{k=1}^{K} \alpha_k \mathbb{1} \left( \widehat{\mathsf{QV}}_k (h) \geq \lambda - \sqrt{\frac{\log \left( \frac{K+2}{\delta} \right)}{2 n_k}} \right)$$

$$\text{subject to} \quad \left| \frac{1}{K} \sum_{k=1}^{K} \alpha_k - 1 \right| \leq c_1 \sqrt{\frac{1}{K} \log \left( K \delta^{-1} \right)},$$

$$\frac{1}{K} \sum_{k=1}^{K} f \left( \alpha_k \right) \leq \varepsilon + c_2 \sqrt{\frac{1}{K} \log \left( K \delta^{-1} \right)},$$

*Then, with probability at least* $1 - \delta$*, the following bound holds uniformly over all* $\lambda \in \mathbb{R}$*:*

$$\sup_{\mu' \in \mathcal{G}_\varepsilon^{f-\mathrm{div}}(\mu)} \mu' \left( \mathbb{E}_P \left[ \ell \left( y, h \left( \boldsymbol{X} \right) \right) \right] \geq \lambda \right) \leq \quad (12)$$

$$\widehat{J}^* (\varepsilon, \lambda) + \mathcal{O} \left( \sqrt{K^{-1} \log \left( K \delta^{-1} \right)} \right).$$

Proof is in Appendix D, and extends the classical results from Glivenko-Cantelli theorem and DKW bound into an adversarial setting. We have detailed our new theoretical findings in Section B. The bound in Theorem 5.3 exhibits the following properties: i) It is forward-shifted with respect to the true CDF, meaning it exhibits a delayed reaction to increasing $\lambda$ compared to the true CDF. The maximum delay is on the order of $\mathcal{O}(\sqrt{\log K / \min_k n_k})$. ii) The value of the CDF estimator also deviates from the true estimator by the amount $\mathcal{O}(\sqrt{\log K / K})$. Consequently, as $K$ and $(\min_k n_k) / \log K$ tend to infinity, the generalization gap becomes zero. iii) The bound holds uniformly over all

$\lambda \in \mathbb{R}$, similar to the original DKW inequality. Again, the empirical value $\widehat{J}^* (\varepsilon, \lambda)$ is the solution to a server-side convex and thus efficient program. More details will be given in Section 7.

## 6. Wasserstein Meta-Distributional Shifts

This section addresses the case of Wasserstein meta-shifts from $\mu$ to $\mu'$. Such shifts present significant challenges, as the model $h$ may encounter entirely unseen regions of the meta-distributional support. In practical terms, users may exhibit data distributions that are entirely novel compared to $\widehat{P}_k$s of the evaluation phase in the source network. To mitigate this effect, as we will show later in this section, server needs to query out-of-domain loss values from each of the $K$ clients. Here, we provide theoretical guarantees for the average risk, but not for the entire risk CDF. Addressing the full risk CDF requires a more advanced methodology, which falls beyond the scope of the current paper and is defered to future work. We begin by introducing both standard and meta-distributional Wasserstein metrics, and then present our main result, an analog of Theorem 5.2 for Wasserstein-type shifts.

**Definition 6.1.** For any two measures $P, Q \in \mathcal{M} \left( \mathcal{Z} \right)$ and a lower semi-continuous function $c : \mathcal{Z} \times \mathcal{Z} \to \mathbb{R}_{\geq 0}$, we define the *Wasserstein* distance between $P$ and $Q$ as

$$\mathcal{W}_c \left( P, Q \right) \triangleq \inf_{\nu \in \mathcal{C}(P,Q)} \mathbb{E}_{(\boldsymbol{Z}, \boldsymbol{Z}') \sim \nu} \left\{ c \left( \boldsymbol{Z}, \boldsymbol{Z}' \right) \right\}, \quad (13)$$

where $\mathcal{C} \left( P, Q \right)$ denotes the set of all couplings between $P$ and $Q$, i.e., all joint probability measures in $\mathcal{M} \left( \mathcal{Z} \times \mathcal{Z} \right)$ that have fixed $P$ and $Q$ as their respective marginals (Kuhn et al., 2019).

The function $c$ is called the *transportation cost* and is user-defined. For example, $c \left( \boldsymbol{Z}, \boldsymbol{Z}' \right) = \| \boldsymbol{X} - \boldsymbol{X}' \|_2 + \infty \cdot \mathbb{1} \left( y \neq y' \right)$ corresponds to a typical *feature-shift* scenario in robust ML. $\mathcal{W}_c \left( P, Q \right)$ (which is a metric over $\mathcal{M}$) measures the minimum cost of transforming $P$ into $Q$ or vice versa according to the cost characterized by $c$. Unlike $f$-divergence, Wasserstein distance can stay bounded under support shifts. In fact, it is a powerful tool to model slight support changes between distributions and due to this property is widely used in adversarial robustness research.

In a similar fashion, one can define the Wasserstein distance between any two meta-distributions $\mu, \mu' \in \mathcal{M}^2 \left( \mathcal{Z} \right)$ with respect to any valid transportation cost over the space of measures $\mathcal{M} \left( \mathcal{Z} \right)$, such as the ordinary Wasserstein distance.

**Definition 6.2** (Wasserstein metric over $\mathcal{M}^2$)**.** For any two meta-distributions $\mu, \mu' \in \mathcal{M}^2 \left( \mathcal{Z} \right)$, assume a distributional transportation cost such as ordinary Wasserstein distance $\mathcal{W}_c \left( \cdot, \cdot \right)$, where $c$ is a bounded and proper (according to Definition 6.2) transportation cost on $\mathcal{Z} \times \mathcal{Z}$. In this regard,

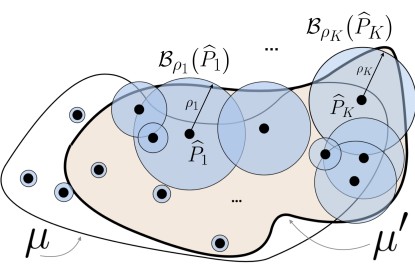

*Figure 3.* A graphical illustration of the core idea behind Theorem 6.3. Each tuple $\rho_{1:K}$ that satisfies the constraints corresponds to one possibility for $\mu'$, via extending the local risk of $\widehat{P}_k$s to their worst-case out-of-distribution values in $\mathcal{B}_{\rho_k}(\widehat{P}_k)$. Therefore, we maximize over all such $\rho_{1:K}$ to account for the worst $\mu'$.

let us define

$$\|\mu - \mu'\|_{\mathcal{W}_c} \triangleq \inf_{\nu \in \mathcal{C}(\mu,\mu')} \mathbb{E}_{(P,Q)\sim\nu}\{\mathcal{W}_c(P,Q)\} \quad (14)$$

as the Wasserstein distance between $\mu$ and $\mu'$ according to transportation cost $\mathcal{W}_c$.

Here, $\mathcal{C}(\mu,\mu')$ is the set of all couplings (joint measures in $\mathcal{M}(\mathcal{M}(\mathcal{Z}) \times \mathcal{M}(\mathcal{Z}))$) with $\mu$ and $\mu'$ as their respective marginals. Accordingly, for $\varepsilon \geq 0$ and $P \in \mathcal{M}$, we define the Wasserstein ball of radius $\varepsilon$ around $P \in \mathcal{M}$ as $\mathcal{B}_{\varepsilon}^{\text{wass}}(P) \triangleq \{Q | \mathcal{W}_c(P,Q) \leq \varepsilon\}$, where the transportation cost $c$ is hidden from formulation for the sake of simplicity. Similarly, one can define the Wasserstein ball around meta-distribution $\mu \in \mathcal{M}^2(\mathcal{Z})$ with radius $\varepsilon$ as $\mathcal{G}_{\varepsilon}(\mu) \triangleq \{\mu' | \|\mu - \mu'\|_{\mathcal{W}_c} \leq \varepsilon\}$, which is the set of meta-distributions with a (meta-distributional) Wasserstein distance of at most $\varepsilon$ from $\mu$.

We now present our main result of this section: a quasi-convex (and thus polynomial-time) optimization problem that provides empirical meta-distributionally robust evaluation guarantees against Wasserstein shifts, with a asymptotically vanishing generalization gap.

**Theorem 6.3** (Empirical Evaluation with Wasserstein Robustness). *Consider the same setting as in Theorem 5.2. Let $c$ be a bounded and proper (according to Definition 6.2) transportation cost on $\mathcal{Z} \times \mathcal{Z}$. For any given $\varepsilon, \delta > 0$, consider the following constrained optimization problem:*

$$\widehat{U}^*(\varepsilon) \triangleq \sup_{\rho_1,\ldots,\rho_K \geq \varepsilon/K} \frac{1}{K}\sum_{k=1}^{K} \widehat{\text{QV}}_k(h,\rho_k) \quad (15)$$

$$\text{subject to} \quad \frac{1}{K}\sum_{k=1}^{K}\rho_k \leq \varepsilon\left(1+\frac{1}{K}\right) + c_1\sqrt{\frac{\log\left(\frac{K+2}{\delta}\right)}{K}},$$

*where $c_1$ is a universal constant. Then, the following bound holds with probability at least $1 - \delta$ for the meta-*

distributionally robust loss of $h$ around $\mu$:

$$\sup_{\mu'\in\mathcal{G}_{\varepsilon}(\mu)} \mathbb{E}_{P\sim\mu'}\left(\mathbb{E}_P\left[\ell\left(y,h\left(\boldsymbol{X}\right)\right)\right]\right) \leq \widehat{U}^*(\varepsilon) +$$

$$\mathcal{O}\left(\sqrt{\frac{1}{K}\log\left(K\delta^{-1}\right)} + \frac{1}{K}\sum_{k=1}^{K}\sqrt{\frac{1}{n_k}\log\frac{Kn_k}{\varepsilon\delta}}\right).$$

The proof is provided in Appendix E. Similar to Theorem 5.2, Theorem 6.3 offers a robust bound on the expected loss under Wasserstein meta-distributional shifts using $\widehat{U}^*(\varepsilon)$. Once again, the generalization gap decreases asymptotically as both $K$ and $\left(\min_{k\in[K]} n_k\right)/\log K$ increase with rate $\mathcal{O}\left(\sqrt{\log K/K} + \max_{k\in[K]}\sqrt{\frac{1}{n_k}\log(Kn_k)}\right)$. It is important to note that the inherent robustness of $h$ against Wasserstein-type distributional shifts, if it exists, would be reflected in $\widetilde{U}^*(\varepsilon)$, thereby reducing the bound.

A notable contribution of our work in this section is that we show robust evaluation against Wasserstein adversarial attacks to $\mu$ is possible using only $\widehat{\text{QV}}_i(h,\rho)$ values. In other words, it is not needed to move data points into a central server to perform collective attacks, and each client can independently assess the performance of $h$ under local attacks. However, the budget values $\rho_k$ for each client need to be carefully selected. In Section 7, we show this is possible at the server-side in polynomial time. Figure 3 gives a graphical illustration of the procedure in Theorem 6.3.

## 7. Asymptotic Minimax Optimality, Privacy, and Computational Efficiency

In this section, we address the computational complexity of our proposed empirical upper-bounds in Theorems 4.1, 5.2, 5.3 and 6.3 both the server and client sides. We also discuss their privacy w.r.t. local user data. Finally, we prove their asymptotic minimax optimality.

*Remark* 7.1 (**Server-side Computational Complexity**). The server-side optimization problems in Theorems 5.2 and 5.3 are convex and efficiently (in polynomial-time w.r.t. $K$) solvable. Each client is queried only once for their *non-robust* (ordinary) loss value ($\rho_k = 0$). Robust local losses are not required for $f$-divergence shifts, since the reweighting attack can be carried out entirely at the server-side. On the other hand, the bound in Theorem 6.3, i.e., $\widehat{U}^*(\varepsilon)$, is the solution to a *quasi-convex* program, where a standard bisection method (e.g., Algorithm 1 in Appendix F), can approximate $\widehat{U}^*(\varepsilon)$ within an arbitrary error margin $\Delta > 0$, in polynomial time w.r.t. $K$ and $\frac{1}{\Delta}$. The total query budget per client in this case is also polynomial with respect to both $K$ and $\log\frac{1}{\Delta}$.

*Remark* 7.2 (**Client-side Computational Complexity**). For the case of $f$-divergence shifts in Theorems 5.2 and 5.3,

clients only return the ordinary loss values which can be computed in time $\mathcal{O}(n_k)$ at client $k$. In Theorem 6.3, assume the transportation cost $c$ is convex with respect to its second argument[2] and is differentiable. Then, the client-side optimization problem to determine $\widehat{\mathsf{QV}}_k(h, \rho)$ in equation 15, for any given $h$ and $\rho$, is convex (Sinha et al., 2018). A standard stochastic gradient descent algorithm can approximate $\widehat{\mathsf{QV}}_k(h, \rho)$ within an arbitrary error margin $\Delta > 0$, with polynomial time complexity relative to $\Delta^{-1}$.

Proofs of Remarks 7.1 and 7.2, as well as the bisection method in Algorithm 1 can be found in Appendix F.

*Remark* 7.3 (**Privacy**). All empirical bounds in this work only use the local ordinary or adversarial loss values, without direct access to local private samples or model gradients. In particular, the empirical value $\widehat{U}^*(\varepsilon)$ in equation 15 relies solely on the Wasserstein adversarial loss of clients, i.e., $\widehat{\mathsf{QV}}_k(h, \rho)$ for polynomially many values of $\rho$. To the best of our knowledge, there are no well-known privacy attacks capable of effectively recovering private data from this procedure. It should be noted that providing Differential Privacy (DP) certificates goes beyond the scope of our work.

Notably, working on novel attacks or providing information-theoretic analysis for the recover-ability of private data from multiple local adversarial loss values can be a proper future research direction. Finally, the following remark (proved in Appendix F) provides tightness guarantees for our bounds.

*Remark* 7.4 (**Asymptotic Minimax Optimality**). All empirical upper-bounds in Theorems 4.1, 5.2, 5.3 and 6.3 are asymptotically minimax optimal, which means they can be achieved by a worst-case shift when $K$ and all $n_k/\log K$ grow toward infinity. This means our bounds are tight and cannot be improved, at least when network and local dataset sizes become sufficiently large.

## 8. Experimental Results

Our work is mainly theoretical; In any case, we present a series of experiments on real-world datasets to show tightness and computability of our bounds in practice. First, we outline our client generation model and present a number of non-robust risk CDF guarantees. A more complete set of experiments with complementary explanations can be found in Appendix G. We simulated a federated learning scenario with $n = 1000$ nodes, where each node contains 1000 local samples. The experiments were conducted using four different datasets: CIFAR-10 (Krizhevsky et al., 2009), SVHN (Netzer et al., 2011), EMNIST (Cohen et al., 2017), and ImageNet (Russakovsky et al., 2015). To create each user's data within the network, we applied three types of affine distribution shifts across users:

---

[2]This assumption is generally not restrictive, as any norm exhibits this property

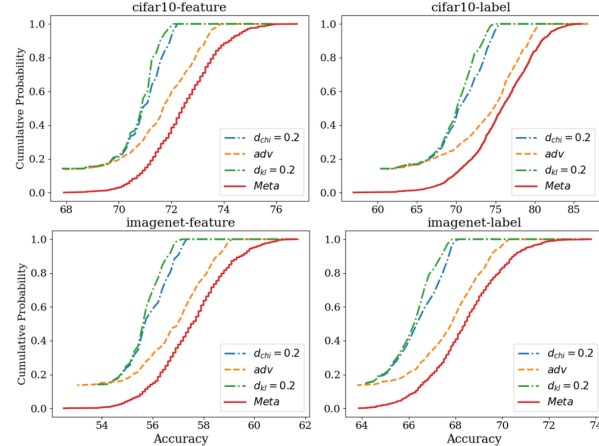

*Figure 4.* Non-robust Risk CDF bounds for unseen clients. Here, "*Meta*" refers to the main population with 1000 nodes. DKW-robust bounds are depicted only for tightness comparison.

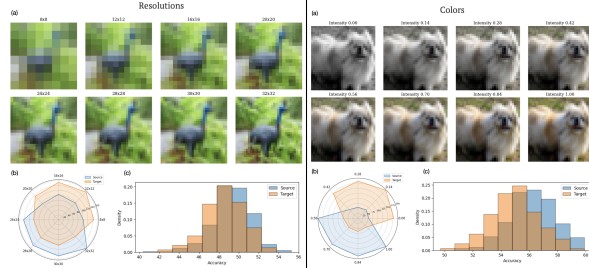

*Figure 5.* Meta-distributional shifts based on resolutions (left) and colors (right). (a) Sample images from each resolution/color. (b) Average number of samples per resolution/color within each network selected from the meta distributions. (c) Histogram of model accuracy densities for the two meta distributions.

**Feature Distribution Shift:** Each sample $X_i^{(k)}$ in the dataset is manipulated via a transformation chosen randomly for each node. Specifically, each user is assigned a unique matrix $\Lambda^{(k)}$ and shift vector $\boldsymbol{\delta}^{(k)}$, and the data is modified as $\tilde{X}_i^{(k)} = (I + \Lambda^{(k)})X_i^{(k)} + \boldsymbol{\delta}^{(k)}$, where $\Lambda^{(k)}$ and $\boldsymbol{\delta}^{(k)}$ are respectively random matrices and vectors with i.i.d. zero-mean Gaussian entries. The standard deviation varies based on the dataset: $0.05$ for CIFAR-10 and SVHN, $0.1$ for EMNIST, and $0.01$ for ImageNet.

**Label Distribution Shift:** Here, we assume that each meta-distribution is characterized by a specific $\alpha$ coefficient. To generate each user's data under this shift, the number of samples per class is determined by a Dirichlet distribution with parameter $\alpha$. In our experiments, we use $\alpha = 0.4$.

**Feature & Label Distribution Shift:** This shift combines both the feature and label distribution shifts described above to create a new distribution for each user.

Figure 4 illustrates our bounds on the risk CDF of unseen

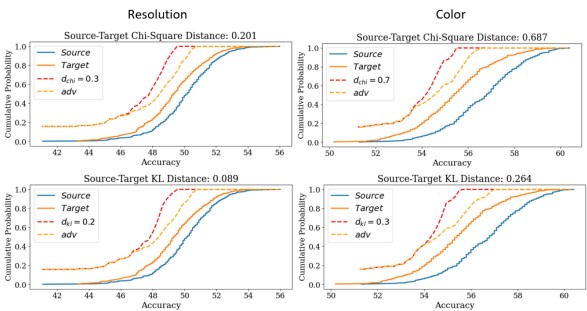

*Figure 6.* f-divergence certificates for two meta-distributions based on resolutions (Left) and colors (Right).

clients with no shifts. We selected 500 nodes from the population and considered 500 other nodes as unseen clients. We then plotted the CDFs based on 500 samples and confirmed that our bounds hold for the real population as well. Due to the standard DKW inequality, the empirical CDF is a good estimate for the test-time non-robust risk CDF.

### 8.1. $f$-Divergence Meta-Distributional Shifts

We assumed users belong to two distinct meta-distributions: the source and the target. A CNN-based model is initially trained on a network of clients sampled from the source. The resulting risk values are then fed into the optimization problems in Section 5 to obtain robust CDF bounds, considering both the Chi-Square and KL divergence as potential choices for $f$. Finally, we empirically estimate the risk CDF for users from the target meta-distribution and validate our bounds. Specifically, we tested our certificates in two distinct settings using the CIFAR-10 dataset (see Figure 5). We generated various image categories with differing resolutions or color schemes, and then sampled from these categories to create different distributions. More details of this procedure can be found in Section G.

Figure 6 verifies our CDF certificates based on both chi-square and KL-divergence (dotted curves) for the target meta distribution (orange curve). As can be seen, bounds have tightly captured the behavior of risk CDF in the target network. More detailed experiments are shown in Figure 9 in Appendix G.

We then used the above-mentioned affine distribution shifts to create new domains according to a Wasserstein metric. Figure 7 summarizes our numerical results in this scenario. To generate different networks within the meta-distribution, we applied the affine distribution shifts described in Section G.1. Once again, the results validate our certificates, this time for Wasserstein-type shifts. It is important to note that the bounds presented here remain tight, particularly under adversarial attacks as defined by a distributional adversary

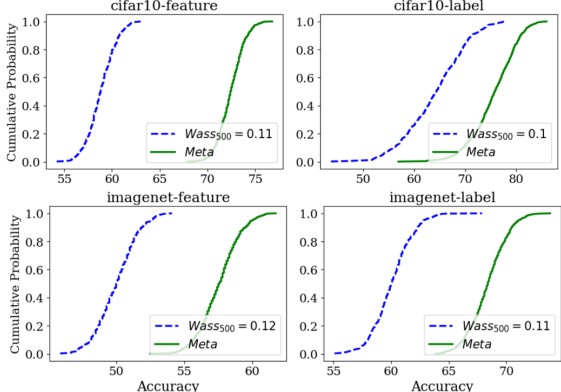

*Figure 7.* Wasserstein-based certificates for unseen clients. "*Meta*" refers to the main population with 1000 nodes. Dotted curves are based on 500 networks within the population.

in (Sinha et al., 2018). More detailed experiments with various levels of tightness are shown in Figure 10 in Appendix G. Although our theoretical findings in Section 6 focus on the average risk and not the risk CDF, we extended the same framework to the CDF in this experiment to explore whether the theory might also apply. The results were positive, suggesting potential for extending our theoretical findings in this area.

## 9. Conclusion

This work introduces new polynomial-time computable performance bounds to guarantee the performance of a model $h$ on an unseen network B, using only data and queries from network A. The key assumption is that the meta-distributions behind user data in the two networks are $\varepsilon$-close, measured by $f$-divergence or Wasserstein shifts, which address diverse practical scenarios. The bounds, backed by rigorous proofs, achieve vanishing generalization gaps as network size $K$ and normalized samples grow. Novel contributions include a robust GC theorem and DKW bound for $f$-divergence shifts. Experiments confirm the bounds' tightness and efficiency.

## Impact Statement

This paper proposes a certifiably robust evaluation framework for federated learning under meta-distributional shifts, providing theoretical guarantees on performance under deployment shifts. The proposed approach may benefit real-world federated systems by improving reliability and trust in evaluation across diverse clients. However, if misapplied without properly validating the assumptions on shift structure, the certificates may give a false sense of robustness. Care should be taken to ensure the theoretical assumptions hold approximately in practice, and to transparently communicate the limitations of the guarantees.

## Acknowledgments

The work of Farzan Farnia is partially supported by a grant from the Research Grants Council of the Hong Kong Special Administrative Region, China, Project 14209920, and is partially supported by CUHK Direct Research Grants with CUHK Project No. 4055164 and 4937054.

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

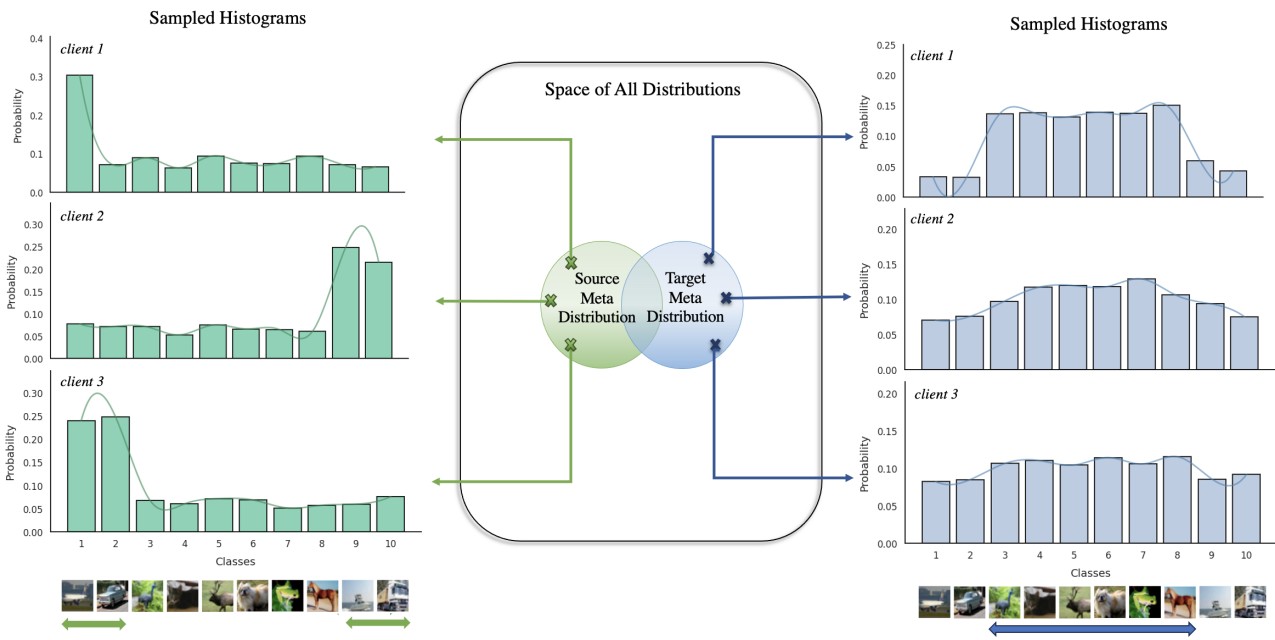

*Figure 8.* A graphical illustration of meta-distributional shifts between two different networks (or societies) of clients/users. Heterogeneous user data distributions can be considered as i.i.d. samples from a meta-distribution. In this example, clients from one meta-distribution primarily have vehicle images, while those from another meta-distribution mostly have animal images.

## A. Complementary Review of Previous works

In this section, we review several related works in federated training and evaluation of models with non-IID clients. We then review a series of known (mostly theoretical) results regrading generalization gaps and shortcomings of current methodologies related to our problem set in this paper, i.e., beta testing in FL scenarios.

### A.1. Generalization in FL

The challenge of generalizing FL models to unseen clients and distributions has been studied in several related works. Li et al. (2020) introduce a method to address the heterogeneity of client data, focusing on improving the generalization of FL models. Similarly, Ma et al. (2024) propose a topology-aware federated learning approach that leverages client relationships to enhance model robustness against out-of-federation data. Also, Zeng et al. (2023) explore adaptive federated learning techniques to dynamically adjust model parameters based on client data distributions. The robustness of FL models against distribution shifts and adversarial attacks has been the focus of several related references. Reisizadeh et al. (2020) propose a robust federated learning framework to handle affine distribution shifts across clients' data. Their proposed framework incorporates a Wasserstein-distance-based distribution shift model to account for device-dependent data perturbations. Also, Zhang et al. (2023) conduct comprehensive evaluations on the adversarial robustness of FL models, proposing the decision boundary-based FL Training algorithm to enhance the the trained model's robustness. Zhou et al. (2022) gain insight from the bias-variance decomposition to improve adversarial robustness in FL. Also, Ben Mansour et al. (2022) propose a robust aggregation method to reduce the effect of adversarial clients.

### A.2. Non-IID Federated Learning

The heterogeneous nature of FL, where clients have different data distributions has been a topic of great interest in the literature. In particular, modeling the heterogeneity of local data distributions/datasets via assuming a higher-level meta-distribution has been the center of several researches.

See, for example, Wu et al. (2024a); Chen et al. (2023); Patel et al. (2022); Matta et al. (2024). Please refer to Figures 1, 2 and 8 for a more illustrative definition of meta-distribution over local statistical or empirical distributions in a heterogeneous and non-IID network. In this regard, Fallah et al. (2020) introduce a personalized federated learning framework based on model-agnostic meta-learning, which provides performance guarantees by optimizing for data distribution heterogeneity. Farnia et al. (2022) propose an optimal transport approach to personalized federated learning by learning and inverting the optimal transport map between the distribution of clients, which has been further extended to fairness-aware personalized federated learning in (Lei et al., 2025). Diamandis et al. (2021) propose a Wasserstein-based framework to federated learning under the non-IID scenario by training a mixed linear regression model capturing the non-IID-ness of the clients' data. Luo et al. (2021) propose a classifier calibration method that adjusts for bias in heterogeneous data, offering improved performance guarantees in non-IID settings. Tan et al. (2022) develop FedProto, a framework that leverages prototype learning to improve convergence and robustness under non-convex objectives. Moreover, Wu et al. (2024b) propose FedLoRA, which adapts low-rank parameter sharing techniques to mitigate the effects of heterogeneity in personalized federated learning. Also, Cheng et al. (2024) and Jia et al. (2024) introduce group-based customization and local parameter sharing strategies, respectively, to provide fairness and efficiency guarantees for heterogeneous client types and multiple tasks in FL. A recent work on using super-quantiles in FL with heterogeneous clients is (Pillutla et al., 2024), which focuses on optimization and convergence aspects of the this problem.

## A.3. Theory of Federated Model Evaluation

Evaluating models over networks of clients, also known as beta testing, or alpha/beta testing, or A/B testing, involves evaluating the performance of a model or application on a small, randomly selected sample of potential users before wider deployment (Soltani et al., 2023). Specifically, this process assumes that a central server has access to a limited subset of $K$ clients from a large network and can request these clients to evaluate a given machine learning model $h$ on their local data. The clients then report the results to the server. The objective is to estimate the *average* performance of the model across the entire network based on this limited sample (Liu et al., 2024).

Several critical challenges emerge in beta testing scenarios: i) **Data Privacy**: Can we reliably assess the average performance without direct access to clients' local data? ii) **Sample Efficiency**: How many randomly selected clients are necessary to achieve a reliable performance assessment? iii) **Meta-Distributional Robustness**: How can we guarantee a minimum level of performance for the model if the underlying meta-distribution governing the clients/users changes slightly? Other considerations include the computational efficiency of server-side and client-side algorithms required for such evaluations.

Questions (i) and (ii) above are relatively well-addressed in the literature, either through federated evaluation frameworks (see Sections A.1 and A.2), or via conventional concentration inequalities (see, for example, our Theorem 4.1). However, the challenge of *Meta-Distributionally Robust Assessment* (M-DRA) remains underexplored. This problem addresses the robustness of model evaluation when the meta-distribution governing client data ($\mu$) shifts slightly to a new distribution ($\mu'$). A straightforward yet naive approach to tackle this problem is to use the concept of **Maximum Mean Discrepancy (MMD)** (see Hu et al. (2024) or Gao et al. (2021)) in order to provide a bound on the performance of $h$ on the target meta-distribution $\mu'$, based on its performance on the source meta-distribution $\mu$ (Sinha et al., 2018). Mathematically, for any meta-distribution $\mu'$ which is in an $\varepsilon$-vicinity of $\mu$, we have:

$$
\begin{aligned}
\mathbb{E}_{P \sim \mu'}\left[\mathbb{E}_P\left(\ell\left(y, h\left(\boldsymbol{X}\right)\right)\right)\right] &= \mathbb{E}_{P \sim \mu}\left[\mathbb{E}_P\left(\ell\left(y, h\left(\boldsymbol{X}\right)\right)\right)\right] + \mathbb{E}_{\mu'-\mu}\left[\mathbb{E}_P\left(\ell\left(y, h\left(\boldsymbol{X}\right)\right)\right)\right] \\
&\leq \mathbb{E}_{P \sim \mu}\left[\mathbb{E}_P\left(\ell\left(y, h\left(\boldsymbol{X}\right)\right)\right)\right] + \sup_{h \in \mathcal{H}} \mathbb{E}_{\mu'-\mu}\left[\mathbb{E}_P\left(\ell\left(y, h\left(\boldsymbol{X}\right)\right)\right)\right] \\
&= \mathbb{E}_{P \sim \mu}\left[\mathbb{E}_P\left(\ell\left(y, h\left(\boldsymbol{X}\right)\right)\right)\right] + \mathrm{MMD}\left(\mu, \mu' | \mathcal{H}\right),
\end{aligned}
\tag{16}
$$

where $\mathcal{H}$ is a class of functions that $h$ belongs to, and $\mathbb{E}_{\mu'-\mu}\left(\cdot\right)$ is defined as $\mathbb{E}_{\mu'}\left(\cdot\right) - \mathbb{E}_{\mu}\left(\cdot\right)$. This way, closeness of $\mu'$ to $\mu$ ($\varepsilon$-vicinity) is reflected into the MMD value, i.e., MMD is proportional to $\varepsilon$. Also, note that the first term, i.e., $\mathbb{E}_{P \sim \mu}\left[\mathbb{E}_P\left(\ell\left(y, h\left(\boldsymbol{X}\right)\right)\right)\right]$ can be estimated according to any unbiased averaging over the sample distributions $P_1, \ldots, P_K$ of $\mu$ (and not $\mu'$), or their empirical counterparts $\widehat{P}_{1:K}$.

While MMD offers a promising starting point, however, the gap term $\mathrm{MMD}\left(\mu, \mu' | \mathcal{H}\right)$ does not shrink as $K$ or either of $n_k$s increase asymptotically. It is an $\varepsilon$-dependent and non-vanishing error term that inflates the adversarial loss of $h$ regardless of any possible inherent robustness in $h$ around $\mu$. To the best of our knowledge, there are no prior works that have already addressed this issue in federated beta testing (or model evaluation in general).

## B. Uniform Convergence of Empirical CDFs from Adversarial Samples: GC Theory Revisited

Assume we are given $n$ i.i.d. samples of a distribution $P$, denoted by $X_1, \ldots, X_n \in \mathbb{R}$ and we aim to estimate the true cumulative distribution function (CDF) of $X \sim P$ based on these empirical observations. The Glivenko-Cantelli theorem provides strong guarantees on the asymptotic behavior of the worst-case error when estimating a CDF from a finite number of i.i.d. samples (Talagrand, 1987). The theorem states that the $\ell_\infty$-norm of the difference between the true CDF of $X \sim P$, denoted by $F$, and its empirical version based on $n$ i.i.d. samples, denoted by $\widehat{F}_n$, which can be formulated as

$$\widehat{F}_n(x) \triangleq \frac{1}{n} \sum_{i \in [n]} \mathbb{1}\left(X_i \leq x\right), \quad \forall x \in \mathbb{R},$$

almost surely converges to zero as $n$ approaches infinity. Mathematically, this is expressed as:

$$\lim_{n \to \infty} \left\| F - \widehat{F}_n \right\|_\infty \overset{a.s.}{=} 0. \tag{17}$$

Furthermore, the well-known Dvoretzky-Keifer-Wolfowitz (DKW) theorem provides non-asymptotic bounds for this asymptotic behavior.

**Lemma B.1** (DKW Inequality (Dvoretzky et al., 1956)). *Let $\mathcal{X}$ be a measurable subset of $\mathbb{R}$, and let $P$ be any probability measure supported on $\mathcal{X}$. Let $\boldsymbol{X}_1, \ldots, \boldsymbol{X}_n \in \mathbb{R}$ be $n$ i.i.d. samples from $P$. Then, the following bound holds for the $\ell_\infty$-norm of the difference between the empirical and true CDF of $P$:*

$$\mathbb{P}\left(\sup_{\lambda \in \mathbb{R}} \left| P\left(\boldsymbol{X} \geq \lambda\right) - \frac{1}{n} \sum_{i \in [n]} \mathbb{1}\left(\boldsymbol{X}_i \geq \lambda\right) \right| \leq \sqrt{\frac{\log \frac{2}{\delta}}{2n}} \right) \geq 1 - \delta, \tag{18}$$

*for any $\delta > 0$.*

The proof of this lemma can be found in the reference. This result allows us to provide uniform convergence guarantees on the tail probability of the loss of $h$, i.e., $\mathbb{P}(\ell\left(y, h\left(\boldsymbol{X}\right)\right) \geq \lambda)$ for any $\lambda \in \mathbb{R}$, based on $K$ i.i.d. observations of the loss across the network. By "uniform," we mean that, with high probability over the sampling of the $K$ users, the worst deviation between the empirical and statistical tail probability is consistently bounded.

For the case of $f$-divergence robustness, we show that it is also possible to derive an asymptotically consistent and *uniform* bound for the risk distribution (CDF) over the *target* network, which estimates

$$\mu'\left(\mathbb{E}_P\left[\ell\left(y, h\left(\boldsymbol{X}\right)\right)\right] \geq \lambda\right),$$

for all $\lambda \in \mathbb{R}$, simultaneously. Here, $\mu'$ represents the unknown target meta-distribution, which is assumed to be within an $\varepsilon$-proximity of the source meta-distribution $\mu$. Note that $\mu$ is also unknown, and our access to it is through $K$ independent empirical realizations $\widehat{P}_1, \ldots, \widehat{P}_K$, where each $\widehat{P}_k$ is known only to client $k$ via a private sample set of size $n_k$. To achieve this, we first derive a robust version of the uniform convergence bound on empirical CDFs, i.e., the robust Dvoretzky-Kiefer-Wolfowitz (DKW) inequality:

**Lemma B.2** (Robust Version of DKW Inequality). *Let $\mu$ and $\mu'$ be two probability measures on $\mathbb{R}$ where $\mu'$ is absolutely continuous w.r.t. $\mu$. Assume $X_1, \ldots, X_n$ for $n \in \mathbb{N}$ to be i.i.d. samples drawn from $\mu$. Suppose we have $\mathcal{D}_f\left(\mu' \| \mu\right) \leq \varepsilon$ for some $\varepsilon \geq 0$ and a proper convex function $f(\cdot)$. For $\delta > 0$, let us define the set $\mathcal{A}_n\left(\varepsilon, \delta\right)$ as*

$$\mathcal{A}_n\left(\varepsilon, \delta\right) \triangleq \left\{ \boldsymbol{\alpha} \in \mathbb{R}^n_+ \left| \left| \frac{1}{n} \sum_{i=1}^n \alpha_i - 1 \right| \leq c_1 \sqrt{n^{-1} \log\left(\frac{1}{\delta}\right)}, \ \frac{1}{n} \sum_{i=1}^n f\left(\alpha_i\right) \leq \varepsilon + c_2 \sqrt{n^{-1} \log\left(\frac{1}{\delta}\right)} \right. \right\}, \tag{19}$$

*where constants $c_1$ and $c_2$ depend only on $f(\cdot)$. Then, there exists $\boldsymbol{\alpha} \in \mathcal{A}_n\left(\varepsilon, \delta\right)$ such that the following uniform bound holds with probability at least $1 - \delta$:*

$$\sup_{\lambda \in \mathbb{R}} \left| \mu'\left(X \leq \lambda\right) - \frac{1}{n} \sum_{i=1}^n \alpha_i \mathbb{1}\left(X_i \leq \lambda\right) \right| \leq \mathcal{O}\left(\sqrt{n^{-1} \log\left(\frac{n}{\delta}\right)}\right). \tag{20}$$

*Proof of Lemma B.2.* The proof for most of its initial parts follows the same path as in the proof of Theorem 5.2. In particular, we use Lemmas D.1 and D.2 from the proof of Theorem 5.2 (see Section D) to show that the following events occur separately with probability at least $1 - \frac{\delta}{3}$, for any $\delta > 0$:

$$\left| \frac{1}{n} \sum_{i \in [n]} \frac{\mathrm{d}\mu'(X_i)}{\mathrm{d}\mu(X_i)} - 1 \right| \leq c_1 \sqrt{\frac{\log \frac{1}{\delta}}{n}}, \tag{21}$$

$$\left| \mathcal{D}_f \left( \mu' \| \mu \right) - \frac{1}{n} \sum_{i \in [n]} f \left( \frac{\mathrm{d}\mu'(X_i)}{\mathrm{d}\mu(X_i)} \right) \right| \leq c_2 \sqrt{\frac{\log \frac{1}{\delta}}{n}}, \tag{22}$$

where $c_1, c_2 > 0$ are constants depending on $f(\cdot)$. These probabilities are with respect to the randomness in drawing i.i.d. samples $X_1, \ldots, X_n \sim \mu$. This is equivalent to the following statement:

$$\mathbb{P} \left[ \left( \frac{\mathrm{d}\mu'(X_i)}{\mathrm{d}\mu(X_i)} \right)_{i \in [n]} \in \mathcal{A}_n \right] \geq 1 - \frac{2\delta}{3}. \tag{23}$$

Next, define $\widehat{F}_n(\lambda)$ for $\lambda \in \mathbb{R}$ as

$$\widehat{F}_n(\lambda) \triangleq \frac{1}{n} \sum_{i=1}^{n} \omega(X_i) \mathbb{1}(X_i \leq \lambda), \tag{24}$$

where the weight function $\omega(X) \geq 0$ is the unknown (bounded) density ratio between $\mu'$ and $\mu$, i.e.,

$$\omega(X) \triangleq \frac{\mathrm{d}\mu'(X)}{\mathrm{d}\mu(X)}. \tag{25}$$

Since $\omega(\cdot)$ is non-negative, $\widehat{F}_n(\lambda)$ is non-decreasing in $\lambda$, starting at 0 when $\lambda = -\infty$ and not exceeding an upper-bound $\Lambda < +\infty$ (due to absolute continuity property) as $\lambda \to \infty$.

Consider the probability measure $\mu'$. For $m \geq 2$, define $\lambda_0^*, \lambda_1^*, \ldots, \lambda_m^*$ such that (i) $\lambda_0^* = -\infty$ and $\lambda_m^* = \infty$, and (ii) $\mu'(X \leq \lambda_i^*) = i/m$ for $i \in [m-1]$. These $\lambda_i^*$, $i \in [m] \cup \{0\}$ represent the $m$-quantiles of $\mu'$. For any $\lambda \in \mathbb{R}$, let $i = i(\lambda) \in [m]$ be such that $\lambda \in [\lambda_{i-1}^*, \lambda_i^*)$. Then, the following chain of inequalities holds almost surely for all $\lambda \in \mathbb{R}$:

$$\widehat{F}_n(\lambda) - \mu'(X \leq \lambda) \leq \widehat{F}_n(\lambda_i^*) - \mu'(X \leq \lambda_{i-1}^*) \leq \widehat{F}_n(\lambda_i^*) - \mu'(X \leq \lambda_i^*) + \frac{1}{m},$$

$$\widehat{F}_n(\lambda) - \mu'(X \leq \lambda) \geq \widehat{F}_n(\lambda_{i-1}^*) - \mu'(X \leq \lambda_i^*) \geq \widehat{F}_n(\lambda_{i-1}^*) - \mu'(X \leq \lambda_{i-1}^*) - \frac{1}{m}. \tag{26}$$

Thus, the following bound holds for all $\lambda \in \mathbb{R}$:

$$\left\| \widehat{F}_n - \mu'(X \leq \cdot) \right\|_\infty = \sup_{\lambda \in \mathbb{R}} \left| \widehat{F}_n(\lambda) - \mu'(X \leq \lambda) \right|$$

$$\leq \max_{i \in \{0, 1, \ldots, m\}} \left| \widehat{F}_n(\lambda_i^*) - \mu'(X \leq \lambda_i^*) \right| + \frac{1}{m}. \tag{27}$$

On the other hand, for any fixed $\lambda \in \mathbb{R}$, we have the following relation for the expectation of $\widehat{F}_n(\lambda)$:

$$\mathbb{E}_\mu[\widehat{F}_n(\lambda)] = \mathbb{E}_\mu \left[ \frac{1}{n} \sum_{i=1}^{n} \omega(X_i) \mathbb{1}(X_i \leq \lambda) \right]$$

$$= \frac{1}{n} \sum_{i=1}^{n} \mathbb{E}_\mu \left[ \omega(X_i) \mathbb{1}(X_i \leq \lambda) \right]$$

$$= \frac{1}{n} \sum_{i=1}^{n} \int_\mathbb{R} \frac{\mathrm{d}\mu'(X)}{\mathrm{d}\mu(X)} \mathbb{1}(X \leq \lambda) \mathrm{d}\mu(X)$$

$$= \mu'(X \leq \lambda). \tag{28}$$

Given that the weight functions $\omega(X_i)$ for $i \in [n]$ are bounded, and $\mathbb{1}(\cdot) \in \{0, 1\}$, McDiarmid's inequality states that for any $\varepsilon > 0$,

$$\mathbb{P}\left(\left|\widehat{F}_n(\lambda_i^*) - \mu'(X \le \lambda_i^*)\right| > \varepsilon\right) \le 2e^{-2\mathcal{O}(n\varepsilon^2)}. \tag{29}$$

Therefore, using the union bound over all $i = 0, 1, \ldots, m$, we obtain:

$$\mathbb{P}\left(\max_{i \in \{0,1,\ldots,m\}}\left|\widehat{F}_n(\lambda_i^*) - \mu'(X \le \lambda_i^*)\right| > \varepsilon\right) \le 2(m+1)e^{-2\mathcal{O}(n\varepsilon^2)}. \tag{30}$$

Equivalently, for any $\delta > 0$, the following bound holds with probability at least $1 - \delta/3$:

$$\max_{i \in \{0,1,\ldots,m\}}\left|\widehat{F}_n(\lambda_i^*) - \mu'(X \le \lambda_i^*)\right| \le \mathcal{O}\left(\sqrt{(2n)^{-1}\log\left(\frac{6(m+1)}{\delta}\right)}\right). \tag{31}$$

Using the preceding inequalities, in particular relations in equation 23, equation 27 and equation 31, we can say there exists $\boldsymbol{\alpha} \in \mathcal{A}_n$ such that the following bounds for $\widehat{F}_n$ hold with probability at least $1 - \delta$:

$$\begin{aligned}
\left\|\widehat{F}_n - \mu'(X \le \cdot)\right\|_\infty &\le \max_{i \in \{0,1,\ldots,m\}}\left|\widehat{F}_n(\lambda_i^*) - \mu'(X \le \lambda_i^*)\right| + \frac{1}{m} \\
&\le \mathcal{O}\left(\inf_{m \in \mathbb{N}_{\ge 2}}\left\{\sqrt{(2n)^{-1}\log\left(\frac{6(m+1)}{\delta}\right)} + \frac{1}{m}\right\}\right) \\
&\le \mathcal{O}\left(\sqrt{n^{-1}\log\left(\frac{n}{\delta}\right)}\right).
\end{aligned} \tag{32}$$

Thus, the proof is complete. $\qquad\square$

A direct corollary of Lemma B.2 is the following robust (again with respect to $f$-divergence adversaries) of the well-known Gilivenko-Cantelli theorem:

**Corollary B.3** (Robust Version of Glivenko-Cantelli Theorem)**.** *Let $\mu$ and $\mu'$ be two probability measures on $\mathbb{R}$ and let $\mathscr{S} \triangleq \{X_i\}_{i=1}^\infty$ be an i.i.d. sequence drawn from $\mu$. Assume $\mu$ and $\mu'$ are absolutely continuous with respect to each other. Additionally, suppose $\mathcal{D}_f(\mu'\|\mu) \le \varepsilon$ for some $\varepsilon \ge 0$ and a proper convex function $f(\cdot)$. Then, there exists a non-negative sequence $\{\alpha_i\}_{i=1}^\infty$ that can depend on $\mathscr{S}$, has the following properties:*

$$\lim_{n\to\infty} \frac{1}{n}\sum_{i=1}^n \alpha_i = 1 \quad \text{and} \quad \lim_{n\to\infty} \frac{1}{n}\sum_{i=1}^n f(\alpha_i) \le \varepsilon, \tag{33}$$

*and also satisfies the following condition:*

$$\lim_{n\to\infty} \sup_{\lambda \in \mathbb{R}} \left|\mu'(X \le \lambda) - \frac{1}{n}\sum_{i=1}^n \alpha_i \mathbb{1}(X_i \le \lambda)\right| \overset{a.s.}{=} 0. \tag{34}$$

Corollary B.3 follows directly from Lemma B.2.

## C. Proofs of the Statements in Section 4

*Proof of Theorem 4.1.* The proof for the average risk is straightforward and combines several applications of McDiarmid's inequality (or, more simply in this case, Hoeffding's inequality). For any instance of distributions $P \sim \mu$, let us define the random variable $\zeta = \zeta(P)$ as follows:

$$\zeta(P) \triangleq \mathbb{E}_P[\ell(y, h(\boldsymbol{X}))] = \int_{\mathcal{Z}} \ell(y, h(\boldsymbol{X})) \, P(\mathrm{d}\boldsymbol{Z}), \tag{35}$$

where $\ell$ denotes the loss function, and $\mathcal{Z}$ is the space of possible outcomes. We omit the detailed proof that $P$ being a random variable implies $\zeta(P)$ is also a random variable, as this follows from standard measurability arguments.

Now, for any $\delta > 0$, we apply McDiarmid's inequality to obtain:

$$\mathbb{P}\left(\zeta - \mathbb{E}_\mu[\zeta] \leq \sqrt{\frac{\log\frac{K+1}{\delta}}{2K}}\right) \geq 1 - \frac{\delta}{K+1}, \tag{36}$$

where the bound holds due to the one-sided version of McDiarmid's inequality, and the fact that $\zeta \in [0, 1]$ almost surely.

Next, for each $k \in [K]$, we similarly have:

$$\mathbb{P}\left(\frac{1}{n_k}\sum_{i=1}^{n_k} \ell\left(y_i^{(k)}, h\left(\boldsymbol{X}_i^{(k)}\right)\right) - \mathbb{E}_{P_i}[\ell(y, h(\boldsymbol{X}))] \leq \sqrt{\frac{\log\frac{K+1}{\delta}}{2n_k}}\right) \geq 1 - \frac{\delta}{K+1}. \tag{37}$$

This follows from Hoeffding's inequality, given that the data points in the local dataset of the $k$-th client are i.i.d. By the union bound, the above $K + 1$ inequalities hold simultaneously with probability at least $1 - \delta$. Finally, combining these inequalities gives us the desired bound in the theorem, thus completing the proof.

For the bound concerning the empirical CDF, the proof follows more or less the same path. Let us define the *statistical query value* of the $k$th client as $\mathsf{QV}_k(h)$, i.e.,

$$\mathsf{QV}_k(h) \triangleq \mathbb{E}_{P_k}[\ell(y, h(\boldsymbol{X}))], \tag{38}$$

where $P_k$ is the true (unknown) data generating distribution which is assigned to client $k \in [K]$. In this regard, according to Lemma B.1, for any $\delta > 0$ we have

$$\mathbb{P}\left(\sup_{\lambda \in \mathbb{R}}\left|\mu\left(\mathbb{E}_P[\ell(y, h(\boldsymbol{X}))] \geq \lambda\right) - \frac{1}{K}\sum_{k \in [K]} \mathbb{1}\left(\mathsf{QV}_k(h) \geq \lambda\right)\right| \leq \sqrt{\frac{\log\frac{2(K+1)}{\delta}}{2K}}\right) \geq 1 - \frac{\delta}{K+1}.$$

Also, applying the McDiarmid's inequality for $K$ times (once, with respect to each client $k \in [K]$), the following bounds also hold:

$$\mathbb{P}\left(\frac{1}{n_k}\sum_{i=1}^{n_k} \ell\left(y_i^{(k)}, h\left(\boldsymbol{X}_i^{(k)}\right)\right) - \mathbb{E}_{P_i}[\ell(y, h(\boldsymbol{X}))] \leq \sqrt{\frac{\log\frac{K+1}{\delta}}{2n_k}}\right) \geq 1 - \frac{\delta}{K+1}, \tag{39}$$

which show the boundedness of the deviation between the empirical query values $\widehat{\mathsf{QV}}_k(h)$ and statistical ones $\mathsf{QV}_k(h)$. This is similar to the previous part of the proof. Again, applying union bound and combining all the inequalities mentioned so far in the proof, the final bound in the statement of the theorem holds with probability at least $1 - \delta$ and the proof is complete. $\qquad\square$

## D. Proofs of the Statements in Section 5

*Proof of Theorem 5.2.* For each $k \in [K]$, let us define the event $\xi_1^{(k)}$ as follows:

$$\xi_1^{(k)} \equiv \mathbb{E}_{P_k}[\ell(y, h(\boldsymbol{X}))] \leq \frac{1}{n_k}\sum_{i \in [n_k]} \ell\left(y_i^{(k)}, h\left(\boldsymbol{X}_i^{(k)}\right)\right) + \sqrt{\frac{\log\left(\frac{K+3}{\delta}\right)}{2n_k}}, \tag{40}$$

where, since $\ell(\cdot)$ is assumed to be a 1-bounded loss function and the samples are drawn independently, McDiarmid's inequality tells us that $\mathbb{P}\left(\xi_1^{(k)}\right) \geq 1 - \frac{\delta}{K+3}$ (McDiarmid et al., 1989). For the rest of the proof, we assume the density ratio of the meta-distributions $\mu, \mu'$ are bounded, i.e., we have

$$\frac{\mathrm{d}\mu'}{\mathrm{d}\mu}(P) \leq \Lambda, \quad \forall P \in \mathrm{supp}(\mu), \tag{41}$$

for some positive $\Lambda$. We call this property the $\Lambda$-*boundedness of density ratio*. At the end of the proof, we show that in our problem we have $\Lambda \leq 1 + \kappa\varepsilon$. We now state and prove two essential lemmas which will be used in the subsequent arguments.

**Lemma D.1.** *Consider two meta-distributions $\mu, \mu' \in \mathcal{M}^2$ which are absolutely continuous with respect to each other and have a $\Lambda$-bounded density ratio for some $\Lambda \geq 1$. For $K \in \mathbb{N}$, assume $P_1, \ldots, P_K \in \mathcal{M}$ to be i.i.d. sample distributions sampled from $\mu$. Then, for all $\epsilon > 0$, the following concentration bound holds:*

$$\mathbb{P}\left(\left|\frac{1}{K}\sum_{k\in[K]}\frac{\mathrm{d}\mu'(P_k)}{\mathrm{d}\mu(P_k)} - 1\right| \geq \epsilon\right) \leq \exp\left(\frac{-2K\epsilon^2}{\Lambda^2\left(1 - \Lambda^{-2}\right)^2}\right). \tag{42}$$

*Proof of Lemma D.1.* Due to the assumed mutual absolute continuity, $\mu$ and $\mu'$ share the same support. Therefore, for $P \sim \mu$, we can define the scalar random variable

$$\zeta = \zeta(P) \triangleq \frac{\mathrm{d}\mu'(P)}{\mathrm{d}\mu(P)}. \tag{43}$$

This variable is bounded by $\Lambda^{-1} \overset{a.s.}{\leq} \zeta \overset{a.s.}{\leq} \Lambda$. Regarding the expected value of $\zeta$, we have:

$$\mathbb{E}\left[\zeta\right] = \mathbb{E}_{P\sim\mu}\left[\frac{\mathrm{d}\mu'(P)}{\mathrm{d}\mu(P)}\right] = \int_{\mathcal{M}}\frac{\mathrm{d}\mu'(P)}{\mathrm{d}\mu(P)}\mathrm{d}\mu(P) = 1. \tag{44}$$

Let $\zeta_k = \zeta(P_k)$. Since $\zeta(P_1), \ldots, \zeta(P_K)$ represent i.i.d. instances of $\zeta$, McDiarmid's inequality states that:

$$\mathbb{P}\left(\left|\frac{1}{K}\sum_{k\in[K]}\zeta_k - \mathbb{E}\left[\zeta\right]\right| \geq \epsilon\right) \leq \exp\left(-\frac{2K\epsilon^2}{\left(\Lambda - \Lambda^{-1}\right)^2}\right), \tag{45}$$

which completes the proof. $\qquad\square$

**Lemma D.2.** *For $\Lambda \geq 1$, assume two meta-distributions $\mu, \mu' \in \mathcal{M}^2$ are absolutely continuous with respect to each other and have a $\Lambda$-bounded density ratio. Let $f$ be a convex function that satisfies the conditions described in Definition 5.1. For $K \in \mathbb{N}$, assume $P_1, \ldots, P_K \in \mathcal{M}$ to be i.i.d. sample distributions sampled from $\mu$. Then, the following concentration bound holds:*

$$\mathbb{P}\left(\left|\mathcal{D}_f\left(\mu'\|\mu\right) - \frac{1}{K}\sum_{k\in[K]}f\left(\frac{\mathrm{d}\mu'\left(P_k\right)}{\mathrm{d}\mu\left(P_k\right)}\right)\right| \geq \epsilon\right) \leq \exp\left(\frac{-2K\epsilon^2}{\mathrm{BW}^2\left(f\left(\cdot\right),\Lambda\right)}\right), \tag{46}$$

*where $\mathrm{BW}\left(f\left(\cdot\right),\Lambda\right)$ is defined as:*

$$\mathrm{BW}\left(f\left(\cdot\right),\Lambda\right) \triangleq \sup_{\Lambda^{-1}\leq u,v\leq\Lambda} f(u) - f(v). \tag{47}$$

*Proof of Lemma D.2.* The proof follows similarly to that of Lemma D.1. For $P \sim \mu$, let us define the random variable

$$\zeta(P) = f\left(\frac{\mathrm{d}\mu'\left(P\right)}{\mathrm{d}\mu\left(P\right)}\right). \tag{48}$$

Then, having defined $\zeta_k = \zeta\left(P_k\right)$, we know that $\zeta_1, \ldots, \zeta_K$ represent i.i.d. instances of $\zeta$. Moreover, the expected value of $\zeta$ is the $f$-divergence between $\mu'$ and $\mu$:

$$\mathbb{E}\left[\zeta\right] = \mathbb{E}_{P\sim\mu}\left[f\left(\frac{\mathrm{d}\mu'(P)}{\mathrm{d}\mu(P)}\right)\right] = \mathcal{D}_f\left(\mu'\|\mu\right). \tag{49}$$

Finally, since $\mu'$ and $\mu$ have the $\Lambda$-bounded density ratio property, the following bounds hold almost surely:

$$\zeta \overset{a.s.}{\leq} \sup_{\Lambda^{-1}\leq u\leq\Lambda} f(u) \quad, \quad \zeta \overset{a.s.}{\geq} \inf_{\Lambda^{-1}\leq v\leq\Lambda} f(v), \tag{50}$$

which means the range of $\zeta$ is almost surely equal to $\mathrm{BW}\left(f\left(\cdot\right), \Lambda\right)$. Hence, again using McDiarmid's inequality, we get the bound:

$$\mathbb{P}\left(\left|\frac{1}{K}\sum_{k\in[K]}\zeta_k - \mathbb{E}\left[\zeta\right]\right| \geq \epsilon\right) \leq \exp\left(\frac{-2K\epsilon^2}{\mathrm{BW}^2\left(f\left(\cdot\right), \Lambda\right)}\right), \tag{51}$$

and this completes the proof. $\qquad\square$

With the lemmas established, let us define additional events $\xi_2$ and $\xi_3$ based on the concentration bounds:

$$\xi_2 \equiv \frac{1}{K}\sum_{k\in[K]}\frac{\mathrm{d}\mu'(P_k)}{\mathrm{d}\mu(P_k)} \lesseqgtr 1 \pm C_1 K^{-1/2}, \tag{52}$$

$$\xi_3 \equiv \frac{1}{K}\sum_{k\in[K]} f\left(\frac{\mathrm{d}\mu'(P_k)}{\mathrm{d}\mu(P_k)}\right) \lesseqgtr \mathcal{D}_f\left(\mu'\|\mu\right) \pm C_2 K^{-1/2}, \tag{53}$$

where $C_1, C_2$ are constants which only depend on $\Lambda$ and $f(\cdot)$, according to Lemmas D.1 and D.2. Based on the above arguments and the results of the mentioned lemmas, we have $\mathbb{P}\left(\xi_2\right), \mathbb{P}\left(\xi_2\right) \geq 1 - \frac{\delta}{K+3}$. Using the central idea for *importance sampling* (Glynn and Iglehart, 1989), the following equations hold for all $\mu, \mu', \ell$ and $h$:

$$\mathbb{E}_{P\sim\mu}\left[\left(\frac{\mathrm{d}\mu'(P)}{\mathrm{d}\mu(P)}\right)\mathbb{E}_P\left[\ell\left(y, h\left(\boldsymbol{X}\right)\right)\right]\right] = \int_{P\in\mathcal{M}}\left(\frac{\mathrm{d}\mu'(P)}{\mathrm{d}\mu(P)}\right)\mathbb{E}_P\left[\ell\left(y, h\left(\boldsymbol{X}\right)\right)\right]\mathrm{d}\mu(P)$$
$$= \mathbb{E}_{P\sim\mu'}\left[\mathbb{E}_P\left[\ell\left(y, h\left(\boldsymbol{X}\right)\right)\right]\right]. \tag{54}$$

At this point, and similar to the idea of Lemma D.1, we define $\xi_4$ as the event of the empirical loss over meta-distribution $\mu'$ concentrates (with high probability) around its expected value, i.e.,

$$\xi_4 \equiv$$
$$\mathbb{E}_{P\sim\mu}\left[\left(\frac{\mathrm{d}\mu'(P)}{\mathrm{d}\mu(P)}\right)\mathbb{E}_P\left[\ell\left(y, h\left(\boldsymbol{X}\right)\right)\right]\right] \leq$$
$$\frac{1}{K}\sum_{k\in[K]}\frac{\mathrm{d}\mu'(P_k)}{\mathrm{d}\mu(P_k)}\mathbb{E}_{P_k}\left[\ell\left(y, h\left(\boldsymbol{X}\right)\right)\right] + \Lambda\sqrt{\frac{\log\left(\frac{K+3}{\delta}\right)}{2K}}. \tag{55}$$

Again, since

$$0 \overset{a.s.}{\leq} \frac{\mathrm{d}\mu'(P_k)}{\mathrm{d}\mu(P_k)}\mathbb{E}_{P_k}\left[\ell\left(y, h\left(\boldsymbol{X}\right)\right)\right] \overset{a.s.}{\leq} \Lambda,$$

McDiarmid's inequality states that the probability bound $\mathbb{P}\left(\xi_4\right) \geq 1 - \frac{\delta}{K+3}$ holds. Our final definition in this proof is a random set of meta-distributions $\mathcal{G} \subseteq \mathcal{M}^2$ which represents an empirical candidate for the neighbors of $\mu$. Mathematically speaking, let us define:

$$\mathcal{G} \triangleq \left\{\nu \in \mathcal{M}^2 \middle| \frac{1}{K}\sum_{k\in[K]}\frac{\mathrm{d}\nu(P_k)}{\mathrm{d}\mu(P_k)} \lesseqgtr 1 \pm C_1 K^{-1/2},\right.$$
$$\left.\frac{1}{K}\sum_{k\in[K]} f\left(\frac{\mathrm{d}\nu\left(P_k\right)}{\mathrm{d}\mu\left(P_k\right)}\right) \leq \varepsilon + C_2 K^{-1/2}\right\}, \tag{56}$$

which depends on $\varepsilon$ and has a random (empirical) nature since it also depends on sample distributions $P_1, \ldots, P_K$. Based on prior discussions and lemmas, we have $\mu' \overset{a.s.}{\in} \mathcal{G}$ as long as the events $\xi_2$ and $\xi_3$ hold, simultaneously. By further assuming

that events $\xi_4$ and $\xi_1^{(k)}$s for all $k \in [K]$ also hold, we can finally write the following chain of inequalities:

$$\mathbb{E}_{P \sim \mu'} \left[ \mathbb{E}_P \left[ \ell \left( y, h \left( \boldsymbol{X} \right) \right) \right] \right] \tag{57}$$

$$\leq \frac{1}{K} \sum_{k \in [K]} \frac{\mathrm{d}\mu'(P_k)}{\mathrm{d}\mu(P_k)} \mathbb{E}_{P_k} \left[ \ell \left( y, h \left( \boldsymbol{X} \right) \right) \right] + \Lambda \sqrt{\frac{\log \left( \frac{K+3}{\delta} \right)}{2K}}$$

$$\leq \frac{1}{K} \sum_{k \in [K]} \frac{\mathrm{d}\mu'(P_k)}{\mathrm{d}\mu(P_k)} \widehat{\mathbb{E}}_{P_k} \left[ \ell \left( y, h \left( \boldsymbol{X} \right) \right) \right] + \Lambda \sqrt{\frac{\log \left( \frac{K+3}{\delta} \right)}{2K}} + \frac{1}{K} \sum_{k \in [K]} \sqrt{\frac{\log \left( \frac{K+3}{\delta} \right)}{2n_k}}$$

$$\overset{a.s.}{\leq} \frac{1}{K} \sup_{\nu \in \mathcal{G}} \sum_{k \in [K]} \frac{\mathrm{d}\nu(P_k)}{\mathrm{d}\mu(P_k)} \widehat{\mathbb{E}}_{P_k} \left[ \ell \left( y, h \left( \boldsymbol{X} \right) \right) \right] + \sqrt{\log \left( \frac{K+3}{\delta} \right)} \left[ \sqrt{\frac{\Lambda^2}{2K}} + \frac{1}{K} \sum_{k \in [K]} \sqrt{\frac{1}{2n_k}} \right].$$

It should be noted that the condition $\nu \in \mathcal{G}$ can be interpreted as introducing

$$\alpha_k \triangleq \frac{\mathrm{d}\nu(P_k)}{\mathrm{d}\mu(P_k)}, \quad \forall k \in [K],$$

and force $\alpha_1 \ldots, \alpha_K$ to satisfy the constraints in the definition of $\widehat{B}^*(\varepsilon)$. Hence, this gives us the high probability bound claimed inside the statement of theorem. The only remaining part of the proof is to show events $\xi_1^{(k)}, \xi_2, \xi_3$ and $\xi_4$ for all $k \in [K]$ hold, simultaneously, with a probability at least $1 - \delta$.

For any event $\xi$, let $\xi^c$ denote its complement. Then, we already have

$$\mathbb{P} \left( \xi_1^{(k)c} \right), \mathbb{P} \left( \xi_2^c \right), \ldots, \mathbb{P} \left( \xi_4^c \right) \leq \frac{\delta}{K+3}, \quad \forall k \in [K].$$

In this regard, one can simply use the union bound and obtain the following chain of inequalities:

$$\mathbb{P} \left( \bigcup_{k \in [K]} \xi_1^{(k)c} \cup \xi_2 \cup \xi_3 \cup \xi_4 \right) \leq \sum_{k \in [K]} \mathbb{P} \left( \xi_1^{(k)c} \right) + \sum_{i=2}^{4} \mathbb{P} \left( \xi_i^c \right) = \frac{K\delta}{K+3} + 3\frac{\delta}{K+3} = \delta. \tag{58}$$

This means the bound in the statement of theorem holds with a probability at least $1 - \delta$, and thus completes the proof.

Finally, let us derive an upper-bound for $\Lambda$, a.k.a., the density ratio bound. According to the definition of $f$-divergence, we have:

$$\mathcal{D}_f \left( \mu' \| \mu \right) = \mathbb{E}_{P \sim \mu} \left[ f \left( \frac{\mathrm{d}\mu'}{\mathrm{d}\mu}(P) \right) \right] \leq \varepsilon. \tag{59}$$

Assume there exists a region in the distributional space $\mathcal{S} \subseteq \mathcal{M}(\mathcal{Z})$, where for all $P \in \mathcal{S}$ the density ratio $\mathrm{d}\mu'/\mathrm{d}\mu(P)$ is at least $\Lambda$, for some $\Lambda \geq 1$. Then, we have $\mathcal{D}_f \left( \mu' \| \mu \right) \geq \mu \left( \mathcal{S} \right) f \left( \Lambda \right)$. Setting $\mu \left( \mathcal{S} \right)$ equal to $\delta$ (our high-probability error margin in this problem), we get

$$f \left( \Lambda \right) \leq \varepsilon / \delta,$$

which means $\Lambda \leq f^{-1} \left( \varepsilon / \delta \right)$. On the other hand, we already know that $f$ is a convex function (see Definition 5.1) which means $f^{-1}$ must be concave. Additionally, we must have $f(1) = 0$. Therefore, the following inequality holds:

$$\Lambda \leq f^{-1} \left( \varepsilon / \delta \right) \leq 1 + \left( \delta^{-1} \left[ f^{-1} \right]' (0) \right) \varepsilon \triangleq 1 + \kappa\varepsilon, \tag{60}$$

where $\kappa$ does not depend on $\varepsilon$. $\qquad \square$

*Proof of Theorem 5.3.* Similar to the proof of Theorem 5.2, we begin by noting that, due to McDiarmid's inequality, for any $\delta > 0$, with probability at least $1 - \frac{K\delta}{K+2}$, the following set of inequalities holds simultaneously for all $k \in [K]$:

$$\widehat{\mathsf{QV}} \left( h \right) \leq \mathsf{QV} \left( h \right) + \sqrt{\frac{\log \left( \frac{K+2}{\delta} \right)}{2n_k}}. \tag{61}$$

Next, it can be readily verified that:

$$\mathbb{1}\left(\mathsf{QV}\left(h\right) \geq \lambda\right) \leq \mathbb{1}\left(\widehat{\mathsf{QV}}\left(h\right) \geq \lambda - \sqrt{\frac{\log\left(\frac{K+2}{\delta}\right)}{2n_k}}\right), \quad \forall k \in [K]. \tag{62}$$

Additionally, note that:

$$\mathbb{E}_{P \sim \mu'}\left[\mathbb{1}\left(\mathsf{QV}\left(h\right) \geq \lambda\right)\right] = \mu'\left(\mathsf{QV}\left(h\right) \geq \lambda\right). \tag{63}$$

The remainder of the proof simply involves applying the result of Lemma B.2 with a maximum error probability of $\frac{2\delta}{K+2}$. This concludes the proof. □

## E. Proofs of the Statements in Section 6

*Proof of Theorem 6.3.* Proof consists of two parts:

- Proving the statement of theorem for the statistical case, where $\min_{k \in [K]} n_k \to \infty$ and thus we have $\widehat{\mathsf{QV}}_k(h, \rho) = \mathsf{QV}_k(h, \rho)$ for all $h \in \mathcal{H}$, $\rho \geq 0$ and $k \in [K]$.

- Replacing the statistically exact adversarial loss $\mathsf{QV}_k(h, \rho)$ which is based on the unknown distribution sample $P_k$ with its empirically calculated counterpart $\widehat{\mathsf{QV}}_k(h, \rho)$ which is computed based on the known (yet private) distribution $\widehat{P}_k$ for all $k \in [K]$. This part of the proof requires establishing a uniform convergence bound over all values of $\rho \geq 0$.

**Part I**  The core mathematical tool used throughout the proof is the following duality result from (Sinha et al., 2018) (originally derived in (Blanchet and Murthy, 2019)) which works for general Wasserstein-constrained optimization problems:

**Lemma E.1** (Proposition 1 of Sinha et al. (2018)). *Let $P$ be a probability measure defined over a measurable space $\Omega$, $\ell(\cdot): \Omega \to \mathbb{R}$ be any loss function, $c$ denote a proper and lower semi-continuous transportation cost on $\Omega \times \Omega$, and assume $\varepsilon \geq 0$. Then, the following equality holds for the Wasserstein-constrained DRO around $P$:*

$$\sup_{Q \in \mathcal{B}_\varepsilon^{\mathrm{wass}}(P)} \mathbb{E}_Q\left[\ell\left(\mathbf{Z}\right)\right] = \inf_{\gamma \geq 0}\left\{\gamma\varepsilon + \mathbb{E}_P\left[\sup_{\mathbf{Z}' \in \Omega} \ell\left(\mathbf{Z}'\right) - \gamma c\left(\mathbf{Z}', \mathbf{Z}\right)\right]\right\}. \tag{64}$$

Proof can be found inside the reference. Also, (Blanchet and Murthy, 2019) and (Zhang et al., 2024) along with several other papers have theoretically analyzed alternative proofs. Based on the duality formulation in Lemma E.1, and considering the fact that meta-distribution $\mu$ is also a "distribution" over the measurable space $\mathcal{M}$, one can rewrite the original Wasserstein-constrained MDRO in the statement of the theorem in its dual form:

$$\sup_{\mu' \in \mathcal{G}_\varepsilon(\mu)} \mathbb{E}_{P \sim \mu'}\left[\mathbb{E}_P\left[\ell\left(y, h\left(\mathbf{X}\right)\right)\right]\right]$$

$$= \inf_{\gamma \geq 0}\left\{\gamma\varepsilon + \mathbb{E}_{P \sim \mu}\left[\sup_Q \mathbb{E}_Q\left[\ell\left(y, h\left(\mathbf{X}\right)\right)\right] - \gamma \mathcal{W}_c\left(P, Q\right)\right]\right\}$$

$$= \inf_{\gamma \geq 0}\left\{\mathbb{E}_\mu\left[\sup_Q \mathbb{E}_Q\left[\ell\left(y, h\left(\mathbf{X}\right)\right)\right] - \gamma\left(\mathcal{W}_c\left(P, Q\right) - \varepsilon\right)\right]\right\}. \tag{65}$$

The main advantage achieved by this reformulation is the substitution of $\mu'$ with the fixed meta-distribution $\mu$ inside the expectation operators. Therefore, the optimization no longer has to be carried out in the $\mathcal{M}^2$ space. For the sake of simplicity in the proof, assume supreme value in equation 65 is attainable. This assumption is not necessary, and can be relaxed by using a more detailed mathematical analysis which is replacing the optimal distribution $Q^*$ with a Cauchy series of distributions and proceed with similar arguments. However, we have decided to avoid this scenario in order to simplify the proof. In this regard, let us define:

$$Q^*\left(P, \gamma; \varepsilon\right) \triangleq \arg\max_Q \mathbb{E}_Q\left[\ell\left(y, h\left(\mathbf{X}\right)\right)\right] - \gamma\left(\mathcal{W}_c\left(P, Q\right) - \varepsilon\right), \quad \forall P \in \mathrm{supp}\left(\mu\right). \tag{66}$$

Then, the following relation holds:

$$\sup_{\mu' \in \mathcal{G}_\varepsilon(\mu)} \mathbb{E}_{P \sim \mu'} \left[ \mathbb{E}_P \left[ \ell\left(y, h\left(\boldsymbol{X}\right)\right) \right] \right] = \inf_{\gamma \geq 0} \left\{ \mathbb{E}_\mu \left[ \mathbb{E}_{Q^*(P, \gamma)} \left[ \ell\left(y, h\left(\boldsymbol{X}\right)\right) \right] - \gamma \left( \mathcal{W}_c\left(P, Q^*\left(P, \gamma\right)\right) - \varepsilon \right) \right] \right\}$$

$$= \inf_{\gamma \geq 0} \left\{ \mathbb{E}_{P \sim \mu} \left[ \mathbb{E}_{Q^*(P, \gamma)} \left[ \ell\left(y, h\left(\boldsymbol{X}\right)\right) \right] \right] - \right.$$
$$\left. \gamma \mathbb{E}_{P \sim \mu} \left[ \mathcal{W}_c\left(P, Q^*\left(P, \gamma\right)\right) - \varepsilon \right] \right\}. \tag{67}$$

which, can be simply rewritten as:

$$\sup_{\mu' \in \mathcal{G}_\varepsilon(\mu)} \mathbb{E}_{P \sim \mu'} \left[ \mathbb{E}_P \left[ \ell\left(y, h\left(\boldsymbol{X}\right)\right) \right] \right] = \inf_{\gamma \geq 0} \mathbb{E}_\mu \left[ \mathbb{E}_{Q^*(P, \gamma)} \left[ \ell\left(y, h\left(\boldsymbol{X}\right)\right) \right] \right]$$

$$\text{subject to} \quad \mathbb{E}_{P \sim \mu} \left[ \mathcal{W}_c\left(P, Q^*\left(P, \gamma\right)\right) \right] \leq \varepsilon. \tag{68}$$

Using a similar argument as before, let us assume the $\inf_{\gamma \geq 0}$ in equation 68 is also attainable and denote the optimal value by $\gamma^* = \gamma^*\left(\mu, \varepsilon\right)$. Once again, this assumption is not necessary and can be relaxed at the expense of introducing more mathematical details and making the proof less readable. In this regard, we have:

$$\sup_{\mu' \in \mathcal{G}_\varepsilon(\mu)} \mathbb{E}_{P \sim \mu'} \left[ \mathbb{E}_P \left[ \ell\left(y, h\left(\boldsymbol{X}\right)\right) \right] \right] = \mathbb{E}_\mu \left[ \mathbb{E}_{Q^*(P, \gamma^*)} \left[ \ell\left(y, h\left(\boldsymbol{X}\right)\right) \right] \right], \tag{69}$$

where it has been already guaranteed that the optimal parameter $\gamma^* \geq 0$ and optimal distribution $Q^*\left(P, \gamma^*\right)$, the following constraint holds:

$$\mathbb{E}_{P \sim \mu} \left[ \mathcal{W}_c\left(P, Q^*\left(P, \gamma^*\right)\right) \right] \leq \varepsilon. \tag{70}$$

For any $P \in \text{supp}\left(\mu\right) \subseteq \mathcal{M}$, let us define the following *optimal robustness radius* function

$$\rho^*\left(P; \varepsilon, \mu\right) \triangleq \mathcal{W}_c\left(P, Q^*\left(P, \gamma^*\right)\right). \tag{71}$$

Therefore, the original MDRO objective in the statement of the theorem can be readily upper-bounded using the following distributionally robust formulation:

$$\sup_{\mu' \in \mathcal{G}_\varepsilon(\mu)} \mathbb{E}_{P \sim \mu'} \left[ \mathbb{E}_P \left[ \ell\left(y, h\left(\boldsymbol{X}\right)\right) \right] \right] = \mathbb{E}_{P \sim \mu} \left[ \mathbb{E}_{Q^*(P, \gamma^*)} \left[ \ell\left(y, h\left(\boldsymbol{X}\right)\right) \right] \right]$$

$$\leq \mathbb{E}_{P \sim \mu} \left[ \sup_{Q \in \mathcal{B}^{\text{wass}}_{\rho^*(P)}(P)} \mathbb{E}_Q \left[ \ell\left(y, h\left(\boldsymbol{X}\right)\right) \right] \right]. \tag{72}$$

Using the upper-bound in equation 72 and the inequality condition on optimal Wasserstein radius functions $\rho^*(P)$ described in equation 70, we can proceed to the empirical stage of the proof. At this stage, the true expectation operators should be replaced by their empirical counterparts which are based on i.i.d. realizations of meta-distribution $\mu$, i.e., unknown distributions $P_1, \ldots, P_K$ and their known yet private empirical realizations, i.e., $\widehat{P}_i$ for $i \in [K]$.

For $P \sim \mu$, let us define the following new and real-valued random variables $\psi(P)$ and $\zeta(P)$ as follows:

$$\psi(P) \triangleq \sup_{Q \in \mathcal{B}^{\text{wass}}_{\rho^*(P)}(P)} \mathbb{E}_Q \left[ \ell\left(y, h\left(\boldsymbol{X}\right)\right) \right],$$

$$\zeta(P) \triangleq \rho^*\left(P; \varepsilon, \mu\right). \tag{73}$$

It should be noted that $\psi(P)$ is readily known to be (almost surely) bounded by 1, since $\ell(\cdot)$ is assumed to be 1-bounded. Additionally, the boundedness for $\zeta(P)$ directly results from the assumption that $c$ is a bounded transportation cost.

**Lemma E.2.** *There exists $R < +\infty$ such that We have $\rho^*(P; \mu, \varepsilon) \overset{a.s.}{<} R$ for $P \sim \mu$.*

*Proof.* The proof is straightforward and directly results from the definition of Wasserstein distance:

$$\zeta(P) \triangleq \rho^*\left(P; \varepsilon, \mu\right) = \mathcal{W}_c\left(P, Q^*\left(P, \gamma^*\right)\right)$$

$$= \inf_{\nu \in \mathcal{C}(P, Q^*)} \mathbb{E}_\nu \left[ c\left(\boldsymbol{Z}, \boldsymbol{Z}'\right) \right]$$

$$\overset{a.s.}{\leq} \sup_{\boldsymbol{Z}, \boldsymbol{Z}' \in \mathcal{Z}} c\left(\boldsymbol{Z}, \boldsymbol{Z}'\right) < +\infty, \tag{74}$$

which concludes the proof. $\square$

Using a similar series of arguments to the ones explained in Lemmas D.1 and D.2 (proof of Theorem 5.2), together with the fact that $\psi(P_k)$s are all bounded by 1, one can directly apply the McDiarmid's inequality and show that the following bound holds with probability at least $1 - \frac{\delta}{K+2}$, for any $\delta > 0$:

$$\mathbb{E}_{P \sim \mu}\left[\sup_{Q \in \mathcal{B}^{\mathrm{wass}}_{\rho^*(P)}(P)} \mathbb{E}_Q\left[\ell\left(y, h\left(\boldsymbol{X}\right)\right)\right]\right] \leq \frac{1}{K} \sum_{k \in [K]} \sup_{Q \in \mathcal{B}^{\mathrm{wass}}_{\rho^*(P_k)}(P_k)} \mathbb{E}_Q\left[\ell\left(y, h\left(\boldsymbol{X}\right)\right)\right] + \sqrt{\frac{\log\left(\frac{K+2}{\delta}\right)}{2K}}. \tag{75}$$

On the other hand, by using the boundedness property for $\zeta(P)$ proved in Lemma E.2 and applying McDiarmid's inequality once again, the following bound holds with probability $1 - \frac{\delta}{K+2}$ (for any $\delta > 0$) for the empirical mean of $\zeta(P)$ over true sample distributions $P_1, \ldots, P_k$:

$$\frac{1}{K} \sum_{k \in [K]} \zeta(P_k) \leq \mathbb{E}_{P \sim \mu}\left[\zeta(P)\right] + c_1 \sqrt{\frac{\log\left(\frac{K+2}{\delta}\right)}{K}} \leq \varepsilon + c_1 \sqrt{\frac{\log\left(\frac{K+2}{\delta}\right)}{K}}, \tag{76}$$

where $c_1$ is a known universal constant depending only on the bound on transportation cost $c$. Here, the last inequality is a direct consequence of the property shown in equation 70.

Let $\mathcal{S} \subset \mathbb{R}^K_{\geq 0}$ be defined as the following subset:

$$\mathcal{S} \triangleq \left\{(\zeta_1, \ldots, \zeta_K) \in \mathbb{R}^K \,\middle|\, \zeta_k \geq 0, \; \forall k \in [K], \; \frac{1}{K} \sum_{k \in [K]} \zeta_k \leq \varepsilon + c_1 \sqrt{\frac{\log\left(\frac{K+2}{\delta}\right)}{K}}\right\}. \tag{77}$$

So far, we have shown that

$$\mathbb{P}\left(\left\{\zeta(P_k)\right\}_{k \in [K]} \in \mathcal{S}\right) \geq 1 - \frac{\delta}{K+2}. \tag{78}$$

In a similar procedure to the one used in the proof of Theorem 5.2, union bound ensures that the bound in equation 75 and the mathematical statement of $\left\{\zeta(P_k)\right\}_{k \in [K]} \in \mathcal{S}$ simultaneously hold with probability at least $1 - \frac{2\delta}{K+2}$. Then the following chain of bounds also hold with the same probability w.r.t. drawing of $P_1, \ldots, P_K$ from $\mu$:

$$\sup_{\mu' \in \mathcal{G}_\varepsilon(\mu)} \mathbb{E}_{P \sim \mu'}\left[\mathbb{E}_P\left[\ell\left(y, h\left(\boldsymbol{X}\right)\right)\right]\right] \leq \mathbb{E}_{P \sim \mu}\left[\sup_{Q \in \mathcal{B}^{\mathrm{wass}}_{\rho^*(P)}(P)} \mathbb{E}_Q\left[\ell\left(y, h\left(\boldsymbol{X}\right)\right)\right]\right]$$

$$\leq \frac{1}{K} \sum_{k \in [K]} \sup_{Q \in \mathcal{B}^{\mathrm{wass}}_{\rho^*(P_k)}(P_k)} \mathbb{E}_Q\left[\ell\left(y, h\left(\boldsymbol{X}\right)\right)\right] + \sqrt{\frac{\log\left(\frac{K+2}{\delta}\right)}{2K}}$$

$$\leq \sup_{\underline{\rho} \in \mathcal{S}} \frac{1}{K} \sum_{k \in [K]} \sup_{Q \in \mathcal{B}^{\mathrm{wass}}_{\rho_k}(P_k)} \mathbb{E}_Q\left[\ell\left(y, h\left(\boldsymbol{X}\right)\right)\right] + \sqrt{\frac{\log\left(\frac{K+2}{\delta}\right)}{2K}}$$

$$= \sup_{\underline{\rho} \in \mathcal{S}} \frac{1}{K} \sum_{k \in [K]} \mathsf{QV}_k\left(h, \rho_k\right) + \sqrt{\frac{\log\left(\frac{K+2}{\delta}\right)}{2K}}. \tag{79}$$

For reasons that become clear in the final stages of the proof, we need to replace the set $\mathcal{S}$ with a new one denoted by $\mathcal{S}'$ which should be defined as:

$$\mathcal{S}' \triangleq \left\{(\zeta_1, \ldots, \zeta_K) \in \mathbb{R}^K \,\middle|\, \zeta_k \geq \frac{\varepsilon}{K}, \; \forall k \in [K], \; \frac{1}{K} \sum_{k \in [K]} \zeta_k \leq \varepsilon\left(1 + \frac{1}{K}\right) + c_1 \sqrt{\frac{\log\left(\frac{K+2}{\delta}\right)}{2K}}\right\}.$$

Evidently, replacing $\mathcal{S}'$ with $\mathcal{S}$ in the maximization step of equation 79, i.e., $\sup_{\underline{\rho} \in \mathcal{S}'}$, gives an upper bound for the original formulation of $\sup_{\underline{\rho} \in \mathcal{S}}$, since each member of $\mathcal{S}'$ can be formed by taking a member from $\mathcal{S}$ and add all radius values by a constant $\varepsilon/K$. Obviously, this procedure only makes the adversarial loss value larger and hence all the bounds still apply.

**Part II:** So far, we have managed to (partially) prove the proposed bound in the statement of the theorem in scenarios where $\min_k \ n_k \to \infty$ and thus we have $\widehat{P}_k \overset{a.s.}{=} P_k$ for all $k \in [K]$. At this stage of the proof we focus on replacing

$$\mathsf{QV}_k\left(h, \rho\right) = \sup_{Q \in \mathcal{B}^{\mathrm{wass}}_{\rho_k}(P_k)} \mathbb{E}_Q\left[\ell\left(y, h\left(\boldsymbol{X}\right)\right)\right],$$

for any $k \in [K]$ and arbitrary $\rho \geq 0$, with its empirical version $\widehat{\mathsf{QV}}_k\left(h, \rho\right)$. Let us reiterate that we do not have any knowledge regarding $P_k$, and only client $k$ has access to its empirical version $\widehat{P}_k$ which is based on $n_k$ i.i.d. samples. Therefore, $\widehat{\mathsf{QV}}_k\left(h, \rho\right)$ is computable via the querying policy described in Section 3, while the true query value $\mathsf{QV}(h, \rho)$ is always unknown.

To this aim, similar to (Sinha et al., 2018) first let us define the following $\phi_\gamma\left(\boldsymbol{Z}\right)$ function for $\gamma \geq 0$ and $\boldsymbol{Z} \in \mathcal{Z}$:

$$\phi_\gamma\left(\boldsymbol{Z}\right) \triangleq \sup_{\boldsymbol{Z}' \in \mathcal{Z}} \ \ell\left(\boldsymbol{Z}'\right) - \gamma c\left(\boldsymbol{Z}', \boldsymbol{Z}\right), \tag{80}$$

where $c(\cdot, \cdot)$ is the original transportation cost and $\ell\left(\boldsymbol{Z}\right)$ abbreviates $\ell\left(y, h\left(\boldsymbol{X}\right)\right)$ where we have omitted $h$ for simplicity in notation. First, it can readily verified that if $\ell$ is bounded between 0 and 1, so does $\phi_\gamma$ for any $\gamma \geq 0$. Second, note that from Lemma E.1 we have the following duality formulation for $\mathsf{QV}_k$ and $\widehat{\mathsf{QV}}_k$ for any $\rho_k \geq 0$:

$$\mathsf{QV}_k\left(h, \rho_k\right) \triangleq \sup_{Q \in \mathcal{B}^{\mathrm{wass}}_{\rho_k}(P_k)} \mathbb{E}_Q\left[\ell\left(y, h\left(\boldsymbol{X}\right)\right)\right] = \inf_{\gamma \geq 0} \left\{\gamma\rho_k + \mathbb{E}_{P_k}\left[\phi_\gamma\left(\boldsymbol{Z}\right)\right]\right\}, \tag{81}$$

$$\widehat{\mathsf{QV}}_k\left(h, \rho_k\right) \triangleq \sup_{Q \in \mathcal{B}^{\mathrm{wass}}_{\rho_k}(\widehat{P}_k)} \mathbb{E}_Q\left[\ell\left(y, h\left(\boldsymbol{X}\right)\right)\right] = \inf_{\gamma \geq 0} \left\{\gamma\rho_k + \frac{1}{n_k}\sum_{i \in [n_k]} \phi_\gamma\left(\boldsymbol{Z}_i^{(k)}\right)\right\}.$$

In the following, first we show that $\phi_\gamma\left(\boldsymbol{Z}\right)$, for any $\boldsymbol{Z} \in \mathcal{Z}$ is a Lipschitz function with respect to $\gamma$ where the Lipschitz constant only depends on the way the transportation cost $c$ is bounded, i.e., the inherent boundedness of $c$ or the compactness of $\mathcal{Z}$. Additionally, we show that the optimal $\gamma \geq 0$ in both minimization problems on the right-hand sides of equation 81 is bounded by a known constant. The latter result is deduced from the fact that all robustness radii $\rho_k$, $k \in [K]$ in the statement of theorem has a known margin from zero. Finally, we show the above-mentioned properties can guarantee that $\mathbb{E}_{\widehat{P}_k}\left[\phi_\gamma\left(\boldsymbol{Z}\right)\right]$ uniformly converges to its true expected value $\mathbb{E}_{P_k}\left[\phi_\gamma\left(\boldsymbol{Z}\right)\right]$ for all relevant value of $\gamma \geq 0$. Hence, the empirical and statistical query values are always within a controlled and asymptotically small deviation from each other regardless of the robustness radius value $\rho_k$.

In this regard, the following lemma shows that $\phi_\gamma\left(\boldsymbol{Z}\right)$ for any $\boldsymbol{Z} \in \mathcal{Z}$ is a Lipschitz function with respect to $\gamma \geq 0$:

**Lemma E.3.** *There exists a constant $R \geq 0$ which only depends on transportation cost $c$ such that function $\phi_\gamma\left(\boldsymbol{Z}\right)$ is $R$-Lipschitz with respect to $\gamma \geq 0$, for all $\boldsymbol{Z} \in \mathcal{Z}$.*

*Proof.* For any two distinct values $\gamma, \gamma' \geq 0$, let $\boldsymbol{Z}_\gamma^*$ and $\boldsymbol{Z}_{\gamma'}^*$ denote the optimal values for which the sup in equation 80 is attained. Similar to several previous arguments, attainability of the sup in this case is not necessary again, and thus this assumption is made for the sake of simplifying the proof.

Then, for any $\boldsymbol{Z} \in \mathcal{Z}$ we have:

$$\phi_\gamma\left(\boldsymbol{Z}\right) = \sup_{\boldsymbol{Z}' \in \mathcal{Z}} \ell\left(\boldsymbol{Z}'\right) - \gamma c\left(\boldsymbol{Z}', \boldsymbol{Z}\right) \geq \ell\left(\boldsymbol{Z}_{\gamma'}^*\right) - \gamma c\left(\boldsymbol{Z}_{\gamma'}^*, \boldsymbol{Z}\right),$$
$$\phi_{\gamma'}\left(\boldsymbol{Z}\right) = \ell\left(\boldsymbol{Z}_{\gamma'}^*\right) - \gamma' c\left(\boldsymbol{Z}_{\gamma'}^*, \boldsymbol{Z}\right),$$

which directly gives us the following bound:

$$\phi_\gamma\left(\boldsymbol{Z}\right) - \phi_{\gamma'}\left(\boldsymbol{Z}\right) \geq -\left(\gamma - \gamma'\right) c\left(\boldsymbol{Z}_{\gamma'}^*, \boldsymbol{Z}\right). \tag{82}$$

Through a set of similar arguments and replacing $\gamma$ and $\gamma'$, the following complementary bound can be achieved as well:

$$\phi_\gamma\left(\boldsymbol{Z}\right) - \phi_{\gamma'}\left(\boldsymbol{Z}\right) \leq -\left(\gamma - \gamma'\right) c\left(\boldsymbol{Z}_\gamma^*, \boldsymbol{Z}\right). \tag{83}$$

Therefore, the following inequality can be established according to the boundedness of $c$ (or alternatively, compactness of $\mathcal{Z}$):

$$
\begin{aligned}
|\phi_\gamma\left(\boldsymbol{Z}\right) - \phi_{\gamma'}\left(\boldsymbol{Z}\right)| &\leq |\gamma - \gamma'| \max_{r \in \{\gamma, \gamma'\}} \{c\left(\boldsymbol{Z}_r^*, \boldsymbol{Z}\right)\} \\
&\leq |\gamma - \gamma'| \sup_{\boldsymbol{Z}' \in \mathcal{Z}} c\left(\boldsymbol{Z}', \boldsymbol{Z}\right) \\
&\leq R|\gamma - \gamma'|,
\end{aligned}
\tag{84}
$$

which proves the Lipschitz-ness of $\phi_\gamma$ with respect to $\gamma$. $\qquad\square$

The following lemma shows that optimal values of $\gamma$ in the right-hand side minimization of equation 81 (or the infimum-achieving sequence in case the infimum is not attainable) is bounded by a known constant:

**Lemma E.4.** *In both minimization problems on the right-hand side of equation 81, the optimal $\gamma$ value denoted by $\gamma^*$ (if attained), or the tail of its sequence in case the $\inf$ is not attainable, satisfies $0 \leq \gamma^* \leq \frac{1}{\rho_k}$.*

*Proof.* Proof directly results from the fact that $\ell(\cdot)$ is bounded between 0 and 1. Therefore, looking at the dual optimization problem in equation 81, increasing $\gamma$ beyond $1/\rho_k$ results in $\gamma\rho_k > 1$ while the second term (i.e., the adversarial loss) is always lower-bounded by zero which makes the whole objective to become larger than 1. On the other hand, setting $\gamma = 0$ would (at worst) results in the objective to be 1. Therefore, the optimizer $\inf_{\gamma \geq 0}$ should not choose a $\gamma$ value that is larger than $1/\rho_k$. $\qquad\square$

At this point, we can state the main lemma in the second part of the proof, which theoretically shows that empirical query values, i.e., $\widehat{\mathsf{QV}}_k(h, \rho)$ for any fixed $h \in \mathcal{H}$ and uniformly all $\rho \geq 0$ converge to their true statistical expected values with a high probability.

**Lemma E.5** (Uniform Convergence of Empirical Queries). *For $k \in [K]$, assume the unknown sample distribution $P_k$ and let $\left\{\boldsymbol{Z}_i^{(k)} = \left(\boldsymbol{X}_i^{(k)}, y_i^{(k)}\right)\right\}$ for $i \in [n_k]$ denote $n_k \in \mathbb{N}$ i.i.d. feature-label pairs drawn from $P_k$. The kth dataset is only known to client $k$. Then, for any fixed classifier $h$ and any $\delta > 0$, the following bound holds with probability at least $1 - \delta/(K+2)$:*

$$
\sup_{\rho \geq \varepsilon/K} \left|\widehat{\mathsf{QV}}_k\left(h, \rho\right) - \mathsf{QV}_k\left(h, \rho\right)\right| \leq \mathcal{O}\left(\sqrt{n_k^{-1} \log\left(\frac{(K+2)n_k}{\varepsilon\delta}\right)}\right).
\tag{85}
$$

*Proof.* Using the dual formulation of Lemma E.1, we can rewrite the main objective of the theorem as follows:

$$
\begin{aligned}
&\sup_{\rho \geq \varepsilon/K} \left|\widehat{\mathsf{QV}}_k\left(h, \rho\right) - \mathsf{QV}_k\left(h, \rho\right)\right| \\
&= \sup_{\rho_k \geq \varepsilon/K} \left\{\inf_{\widehat{\gamma} \geq 0} \left[\widehat{\gamma}\rho_k + \mathbb{E}_{\widehat{P}_k}\left[\phi_{\widehat{\gamma}}\left(\boldsymbol{Z}\right)\right]\right] - \inf_{\gamma \geq 0} \left[\gamma\rho_k + \mathbb{E}_{P_k}\left[\phi_\gamma\left(\boldsymbol{Z}\right)\right]\right]\right\}.
\end{aligned}
\tag{86}
$$

Again, for the sake of simplicity in the proof let us assume both optimal values $\gamma^*$ and $\widehat{\gamma}^*$ in the minimization problems on the right-hand side of equation 86 are attainable. It should be noted that this assumption is not necessary and can be relaxed by adding more mathematical work. Then, from Lemma E.4 we already know

$$
0 \leq \gamma^*, \widehat{\gamma}^* \leq \frac{1}{\rho_k} \leq \frac{K}{\varepsilon}.
$$

Let us partition the feasible search set of $\gamma, \widehat{\gamma} \geq 0$, i.e., $[0, K/\varepsilon]$ into $L \triangleq \lceil\frac{K^2 R}{\varepsilon\Delta}\rceil$ equal intervals, where $\Delta > 0$ is a small constant which should to be determined later in the proof. For each interval, let us choose a representative (for example, the value at the beginning of the interval) denoted by $\gamma_i$ with $i = 1, \ldots, L$. Then, based on Lemma E.3, for any $\rho_k \in [\varepsilon/K, 2K\varepsilon]$, any corresponding $\gamma \in [0, 1/\rho_k]$ and all $\boldsymbol{Z} \in \mathcal{Z}$ we have

$$
|\phi_\gamma\left(\boldsymbol{Z}\right) - \phi_{\gamma_{i^*}}\left(\boldsymbol{Z}\right)| \leq R|\gamma - \gamma_{i^*}| \leq R \cdot \frac{K}{\varepsilon} \cdot \frac{\varepsilon\Delta}{K^2 R} = \frac{\Delta}{K},
\tag{87}
$$

where $i^* = \arg\min_{i \in [L]} |\gamma - \gamma_i|$. On the other hand, from the maximization problem that defines $\widehat{U}^*(\varepsilon)$ in equation 15, we have that

$$\rho_k \leq \mathcal{O}\left(K\varepsilon + \sqrt{K \log\left(\frac{K+2}{\delta}\right)}\right), \ \forall k \in [K], \tag{88}$$

where we have omitted constants for the sake of readability. Therefore, the above discussions directly lead to the following bound in the statistical sense (i.e., with respect to $P_k$):

$$\left| \inf_{\gamma \geq 0} \ [\gamma \rho_k + \mathbb{E}_{P_k} [\phi_\gamma (\boldsymbol{Z})]] - \min_{i \in [L]} \ [\gamma_i \rho_k + \mathbb{E}_{P_k} [\phi_{\gamma_i} (\boldsymbol{Z})]] \right|$$

$$\leq \ \min_{i \in [L]} |\gamma^* - \gamma_i| (R + \sup \ \rho_k)$$

$$\leq \ \frac{K}{\varepsilon} \cdot \frac{\varepsilon \Delta}{K^2 R} \cdot \mathcal{O}\left(R + K\varepsilon + \sqrt{K \log\left(\frac{(K+2)}{\delta}\right)}\right)$$

$$\leq \ \Delta \cdot \mathcal{O}\left(\frac{1}{K} + \frac{\varepsilon}{R} + \frac{1}{R}\sqrt{\frac{\log\left(\frac{K+2}{\delta}\right)}{K}}\right). \tag{89}$$

Through a similar procedure, the following bound also holds for the empirical case (i.e., $\widehat{P}_k$), but this time *almost surely*:

$$\left| \inf_{\widehat{\gamma} \geq 0} \ \left[\widehat{\gamma} \rho_k + \mathbb{E}_{\widehat{P}_k} [\phi_{\widehat{\gamma}} (\boldsymbol{Z})]\right] - \min_{i \in [L]} \ \left[\gamma_i \rho_k + \mathbb{E}_{\widehat{P}_k} [\phi_{\gamma_i} (\boldsymbol{Z})]\right] \right|$$

$$\stackrel{a.s.}{\leq} \ \Delta \cdot \mathcal{O}\left(\frac{1}{K} + \frac{\varepsilon}{R} + \frac{1}{R}\sqrt{\frac{\log\left(\frac{K+2}{\delta}\right)}{K}}\right). \tag{90}$$

Therefore, according to the fact that $\widehat{P}_k$ is an empirical estimate of $P_k$ based on $n_k$ i.i.d. samples , we have:

$$\sup_{\rho \geq \varepsilon/K} \ \left|\widehat{\mathrm{QV}}_k (h, \rho) - \mathrm{QV}_k (h, \rho)\right| \stackrel{a.s.}{\leq} \Delta \cdot \mathcal{O}\left(\frac{1}{K} + \frac{\varepsilon}{R} + \frac{1}{R}\sqrt{\frac{\log\left(\frac{K+2}{\delta}\right)}{K}}\right) +$$

$$\max_{i \in [L]} \left|\mathbb{E}_{\widehat{P}_k} [\phi_{\gamma_i} (\boldsymbol{Z})] - \mathbb{E}_{P_k} [\phi_{\gamma_i} (\boldsymbol{Z})]\right|. \tag{91}$$

Since $\phi_{\gamma_i}$ for each $i \in [L]$ is a non-negative and 1-bounded (adversarial) loss function, simply applying McDiarmid's inequality and the union bound over all $L$ values of $\gamma_i$s would give us the following bound which holds with probability at least $1 - \delta/(K+2)$ for any $\delta > 0$:

$$\max_{i \in [L]} \left|\mathbb{E}_{\widehat{P}_k} [\phi_{\gamma_i} (\boldsymbol{Z})] - \mathbb{E}_{P_k} [\phi_{\gamma_i} (\boldsymbol{Z})]\right| \leq \sqrt{\frac{\log\left[\frac{L(K+2)}{\delta}\right]}{2n_k}}, \tag{92}$$

which gives us the following bound for $\sup_{\rho \geq \varepsilon/K} \left|\widehat{\mathrm{QV}}_k (h, \rho) - \mathrm{QV}_k (h, \rho)\right|$ with the same high probability:

$$\sup_{\rho \geq \varepsilon/K} \ \left|\widehat{\mathrm{QV}}_k (h, \rho) - \mathrm{QV}_k (h, \rho)\right| \leq \mathcal{O}\left(\inf_{\Delta > 0} \left\{\Delta \zeta (K, \varepsilon, \delta) + \sqrt{\frac{\log\left[\frac{RK^2(K+2)}{\varepsilon \delta \Delta}\right]}{n_k}}\right\}\right), \tag{93}$$

where function $\zeta(\cdot)$ is defined as

$$\zeta(K, \varepsilon, \delta) \triangleq \frac{1}{K} + \frac{\varepsilon}{R} + \frac{1}{R}\sqrt{\frac{\log\left(\frac{K+2}{\delta}\right)}{K}}.$$

It should be noted that $R$ is a constant that does not depend on other parameters. Also, we have minimized over $\Delta > 0$ since the bound holds irrespective of $\Delta$. Exact solution of the minimization problem in equation 93 is not needed, since choosing $\Delta = \mathcal{O}\left(K^{-1}n_k^{-1/2}\right)$ gives us the following bound:

$$\sup_{\rho \geq \varepsilon/K} \left|\widehat{\mathsf{QV}}_k\left(h, \rho\right) - \mathsf{QV}_k\left(h, \rho\right)\right| \leq \mathcal{O}\left(\sqrt{\frac{\log\left(\frac{(K+2)n_k}{\varepsilon\delta}\right)}{n_k}}\right), \tag{94}$$

and completes the proof. $\qquad\square$

By using the uniform convergence result from Lemma E.5 and applying it to all $K$ clients simultaneously, we see that (via a union bound argument) with probability at least $1 - K\delta/(K+2)$ the empirical and statistical queries $\widehat{\mathsf{QV}}_k\left(h, \rho_k\right)$ and $\mathsf{QV}_k\left(h, \rho_k\right)$ are asymptotically close for all $k \in [K]$, and uniformly for all robustness radii

$$\rho_k \in \left[\frac{\varepsilon}{K}, \mathcal{O}\left(K\varepsilon + \sqrt{\frac{\log((K+2)/\delta)}{K}}\right)\right]$$

which are considered for the maximization problem of equation 15. Finally, using another union bound argument to incorporate the bounds from equation 75 and equation 78 in addition to the previous $K$ events, we can say that with probability at least $1 - \delta$ the following bound holds:

$$\sup_{\mu' \in \mathcal{G}_\varepsilon(\mu)} \mathbb{E}_{P \sim \mu'}\left[\mathbb{E}_P\left[\ell\left(y, h\left(\boldsymbol{X}\right)\right)\right]\right]$$

$$\leq \sup_{\underline{\rho} \in \mathcal{S}'} \frac{1}{K} \sum_{k \in [K]} \sup_{Q \in \mathcal{B}_{\rho_k}^{\mathrm{wass}}(P_k)} \mathbb{E}_Q\left[\ell\left(y, h\left(\boldsymbol{X}\right)\right)\right] + c_2\sqrt{\frac{\log\left(\frac{(K+2)n_k}{\varepsilon\delta}\right)}{n_k}} + \sqrt{\frac{\log\left(\frac{K+2}{\delta}\right)}{2K}}$$

$$= \sup_{\underline{\rho} \in \mathcal{S}'} \frac{1}{K} \sum_{k \in [K]} \widehat{\mathsf{QV}}_k\left(h, \rho_k\right) + c_2\sqrt{\frac{\log\left(\frac{(K+2)n_k}{\varepsilon\delta}\right)}{n_k}} + \sqrt{\frac{\log\left(\frac{K+2}{\delta}\right)}{2K}}, \tag{95}$$

where $c_2$ is a constant that only depends on either the transportation cost $c$ or the compactness of sample space $\mathcal{Z}$. This completes the proof. $\qquad\square$

# F. Proofs of Remarks in Section 7

*Proof of Remark 7.1.* We begin by discussing the server-side programs in Section 5, which address $f$-divergence meta-robustness, followed by the analysis in Section 6, which focuses on Wasserstein meta-robustness.

**Server-side Programs in Theorems 5.2 and 5.3** : The objective function in both programs is a linear function of the optimization variables $\alpha_1, \ldots, \alpha_K$. Furthermore, the constraints define a convex feasible set:

- The constraint

$$\left|\frac{1}{K}\sum_{k=1}^{K} \alpha_k - 1\right| \leq C,$$

for any positive constant $C$, is equivalent to the intersection of the two constraints: $\frac{1}{K}\sum_{k=1}^{K} \alpha_k \leq 1 + C$ and $\frac{1}{K}\sum_{k=1}^{K} \alpha_k \geq 1 - C$. These correspond to a band region bounded by two hyperplanes in $\mathbb{R}^K$, forming a convex set.

- The other constraint,

$$\frac{1}{K}\sum_{k=1}^{K} f(\alpha_k) \leq \varepsilon + C',$$

---

**Algorithm 1** Server-side Bisection Algorithm

---

**Require:** $K, \varepsilon, \delta$

**input** $h, \Delta > 0$, and $\text{poly}\left(K, \log \frac{1}{\Delta}\right)$ query budget for $\widehat{\mathsf{QV}}_k$, for all $k \in [K]$

 1: Initialize $a \leftarrow \min \ell(\cdot)$ (or 0), $b \leftarrow \max \ell(\cdot)$ (or 1)
 2: **while** $b - a > \Delta$ **do**
 3:     $t \leftarrow (a + b)/2$
 4:     Solve convex feasibility problem:
 5:     Find $\rho_1, \dots, \rho_K \geq \varepsilon/K$ such that
 6:        $\frac{1}{K} \sum_{k \in [K]} \rho_k \leq \varepsilon \left(1 + \frac{1}{K}\right) + c_1 \sqrt{\frac{\log\left(\frac{K+2}{\delta}\right)}{K}}$
 7:        $\frac{1}{K} \sum_{k \in [K]} \widehat{\mathsf{QV}}_k(h, \rho_k) \geq t$
 8:     **if** problem is feasible **then**
 9:        $a \leftarrow t$
10:     **else**
11:        $b \leftarrow t$
12:     **end if**
13: **end while**

**output** upper-bound $b$

---

for a constant $C'$, represents the $(\varepsilon + C')$-sublevel set of the convex function $\frac{1}{K} \sum_{k=1}^{K} f(\alpha_k)$. The convexity of this function follows directly from the convexity of $f(\cdot)$, as stated in Definition 5.1. Consequently, this constraint also defines a convex set.

Therefore, both programs feature linear objectives and feasible sets that are intersections of convex sets in $\mathbb{R}^K$, implying that both programs are convex and can be solved efficiently (Boyd, 2004; Ghadimi and Lan, 2013). This completes the proof.

**Server-side Program in Theorem 6.3** : The query functions $\widehat{\mathsf{QV}}_k(h, \rho)$, for any fixed $h \in \mathcal{H}$, are non-decreasing with respect to $\rho \geq 0$. Therefore, the following function:

$$\zeta\left(\rho_1, \dots, \rho_K\right) \triangleq \frac{1}{K} \sum_{k \in [K]} \widehat{\mathsf{QV}}_k\left(h, \rho_k\right) \tag{96}$$

is a summation of $K$ non-decreasing functions, where each function depends only on one of the $\rho_k$ values. This function, $\zeta(\rho_1, \dots, \rho_K)$, is a *quasi-concave* function. Although quasi-concave functions are not generally concave, they do possess concave superlevel sets. Specifically, for each $t \in \mathbb{R}$, the following sets are convex in $\mathbb{R}^K$:

$$\mathcal{S}_t \triangleq \left\{ (\rho_1, \dots, \rho_K) \in \mathbb{R}^K \,\middle|\, \frac{1}{K} \sum_{k \in [K]} \widehat{\mathsf{QV}}_k\left(h, \rho_k\right) \geq t \right\}. \tag{97}$$

As a result, the original optimization problem in equation 15 can be decomposed into a sequence of convex optimization problems. Each sub-problem is essentially a feasibility problem that checks whether the set $\mathcal{S}_t$ is feasible for a given $t \in \mathbb{R}$. Given that the original objective function is bounded between 0 and 1, a binary search can be employed to iteratively approximate the maximum attainable value of the objective within any desired error margin $\Delta > 0$. This process is detailed in Algorithm 1, which implements a *bisection* algorithm.

It is important to note that each feasibility check sub-problem in Algorithm 1 is a convex problem, and therefore can be solved in polynomial time with polynomial evaluations of the constraint functions, i.e., the $\widehat{\mathsf{QV}}_k$ functions. Consequently, both the computational complexity and the required query budget remain polynomial. In particular, for a given $\Delta > 0$, the maximum of the objective in equation 96 can be approximated with an error of at most $\Delta$, requiring at most $\log \frac{1}{\Delta}$ iterations (feasibility checks) in Algorithm 1 (Boyd, 2004). $\qquad \square$

*Proof of Remark 7.2.* We turn to client-side optimizations for this theorem, which take the form:

$$\sup_{Q \in \mathcal{B}_\rho^{\text{wass}}(P)} \mathbb{E}_Q \left[\ell\left(\boldsymbol{Z}\right)\right],$$

where the notation $\ell\left(y, h\left(\boldsymbol{X}\right)\right)$ has been simplified to $\ell\left(\boldsymbol{Z}\right)$ (for $\boldsymbol{Z} = (\boldsymbol{X}, y)$) for clarity and ease of notation. For a client $k \in [K]$, the local distribution $P$ corresponds to $\widehat{P}_k$, which is derived from $n_k$ samples drawn from an arbitrary distribution $P_k$. This optimization program is initially defined within a restricted distributional space, $\mathcal{B}_\rho^{\text{wass}}$, and may appear intractable at first glance. However, by leveraging Lemma E.1 (see Appendix E), which relies on a core mathematical tool from the seminal works of (Sinha et al., 2018) (originally derived in (Blanchet and Murthy, 2019)), this formulation can be rewritten in its dual form. This dual reformulation is subsequently shown to be implementable in polynomial time (Sinha et al., 2018). The rest of the proof of Remark 7.2 is the summary of the method proposed and subsequently used in the above-mentioned reference (**not our work**).

Let $P$ be a probability measure defined over a measurable space $\Omega$, $\ell(\cdot) : \Omega \to \mathbb{R}$ be any loss function, $c$ denote a proper and lower semi-continuous transportation cost on $\Omega \times \Omega$, and assume $\varepsilon \geq 0$. Then, the following equality holds for the Wasserstein-constrained DRO centered around $P$:

$$\sup_{Q \in \mathcal{B}_\rho^{\text{wass}}(P)} \mathbb{E}_Q \left[\ell\left(\boldsymbol{Z}\right)\right] = \inf_{\gamma \geq 0} \left\{\gamma\rho + \mathbb{E}_P \left[\sup_{\boldsymbol{Z}' \in \Omega} \ell\left(\boldsymbol{Z}'\right) - \gamma c\left(\boldsymbol{Z}', \boldsymbol{Z}\right)\right]\right\}. \tag{98}$$

The optimization over $\gamma$ is one-dimensional. On the other hand, the term $\mathbb{E}P\left[\sup \boldsymbol{Z}' \in \Omega \ell\left(\boldsymbol{Z}'\right) - \gamma c\left(\boldsymbol{Z}', \boldsymbol{Z}\right)\right]$ is non-increasing for $\gamma \geq 0$, while the other term, $\gamma\rho$, is affine. In fact, $\gamma$ and $\rho$ are *dual* counterparts, where increasing $\rho$ decreases the optimal $\gamma$ value and vice versa. Therefore, the outer minimization $\inf_{\gamma \geq 0}$ can be handled in polynomial time as long as the inner expected maximization can be numerically solved efficiently (in polynomial time).

For the inner maximization, we have

$$\mathbb{E}_{\widehat{P}_k} \left[\sup_{\boldsymbol{Z}' \in \Omega} \ell\left(\boldsymbol{Z}'\right) - \gamma c\left(\boldsymbol{Z}', \boldsymbol{Z}\right)\right] = \frac{1}{n_k} \sum_{i \in [n_k]} \sup_{\boldsymbol{Z}' \in \Omega} \ell\left(\boldsymbol{Z}'\right) - \gamma c\left(\boldsymbol{Z}', \boldsymbol{Z}_i\right),$$

for $\boldsymbol{Z}_i = \left(\boldsymbol{X}_i^{(k)}, y_i^{(k)}\right)$, which follows from the fact that $\widehat{P}_k$ is an atomic *empirical* measure. We follow the same procedure as in (Sinha et al., 2018):

- Assume $\ell$ is twice differentiable with a bounded-from-below (but potentially negative) curvature, i.e., the Hessian matrix satisfies $\nabla^2 \ell \succeq -\beta \boldsymbol{I}$ for some bounded $\beta \geq 0$.

- Assume $c$ is 1-strongly convex in its first argument (same as Assumption A in Sinha et al. (2018)).

These assumptions are not overly restrictive, as most candidates for the loss function $\ell(\cdot)$, such as cross-entropy loss coupled with a deep neural network architecture for $h$, directly correspond to analytic and twice-differentiable functions with bounded curvature from below. Note that we **do not** assume convexity for $\ell$, thus maintaining the generality of the problem. On the other hand, most choices for the transportation cost $c$ are convex in their first argument (e.g., any valid norm).

Based on the above-mentioned two assumptions, it is straightforward to see that the term $\ell\left(\boldsymbol{Z}'\right) - \gamma c\left(\boldsymbol{Z}', \boldsymbol{Z}_i\right)$ is *concave* as long as $\gamma \geq \beta$ (i.e., $\varepsilon$ is not excessively large). According to standard convergence theorems in convex optimization theory (see, for example, Ghadimi and Lan (2013) or Boyd (2004)), any vanilla local first-order optimization algorithm with oracle access to gradients, such as stochastic gradient descent (SGD), can achieve an $\mathcal{O}\left(T^{-1/2}\right)$ approximation of

$$\text{Obj}\left(\boldsymbol{Z}_1, \ldots, \boldsymbol{Z}_{n_k}; \gamma\right) \triangleq \frac{1}{n_k} \sum_{i \in [n_k]} \sup_{\boldsymbol{Z}' \in \Omega} \ell\left(\boldsymbol{Z}'\right) - \gamma c\left(\boldsymbol{Z}', \boldsymbol{Z}_i\right),$$

after $T \geq 2$ iterations. This completes the proof. $\qquad\square$

*Proof of Remark 7.4.* As $K, \min_k n_k \to \infty$, the bounds presented in Theorems 4.1, 5.2, 5.3, and 6.3 achieve minimax optimality, referred to as Asymptotic Minimax Optimality (AMO). Mathematically, this implies that the proposed empirical bounds on the right-hand sides become attainable by an adversary, rendering them unimprovable. This particular approach in demonstrating the tightness of such bounds is a standard practice in nearly all theoretical works within this field and related areas (see Sinha et al. (2018); Ashtiani et al. (2018); Saberi et al. (2023); etc.).

**AMO for Theorem 4.1** : This theorem provides non-robust guarantees and thus there are no adversaries involved. Therefore, the proof for this part is straightforward since we only need to show the right-hand side of the inequalities equal the left-hand sides in the asymptotic regime. In this regard, we have

$$
\lim_{K,n_{1:K}\to\infty} \frac{1}{K}\sum_{k=1}^{K} \widehat{\mathsf{QV}}_k(h) = \lim_{K\to\infty} \frac{1}{K}\sum_{k=1}^{K}\left(\lim_{n_k\to\infty}\frac{1}{n_k}\sum_{i=1}^{n_k}\ell\left(y_i^{(k)}, h\left(\boldsymbol{X}_i^{(k)}\right)\right)\right)
$$
$$
\overset{a.s.}{=} \lim_{K\to\infty}\frac{1}{K}\sum_{k=1}^{K}\mathbb{E}_{P_k}\left[\ell\left(y, h\left(\boldsymbol{X}\right)\right)\right]
$$
$$
\overset{a.s.}{=} \mathbb{E}_{P\sim\mu}\left(\mathbb{E}_{P_k}\left[\ell\left(y, h\left(\boldsymbol{X}\right)\right)\right]\right). \tag{99}
$$

The first *almost sure* equality follows from the fact that that dataset $\left\{\left(\boldsymbol{X}_i^{(k)}, y_i^{(k)}\right)\right\}_{i=1}^{n_k}$ consists of i.i.d. samples from $P_k$ for each $k \in [K]$. The second one is a a result of $P_1, \ldots, P_K$ being i.i.d. samples of $\mu$. In both cases we have used the strong law of large numbers.

In a similar fashion, for the loss CDF and all subsequent values of $\lambda \in \mathbb{R}$, the following almost sure equalities hold:

$$
\lim_{K,n_{1:K}\to\infty}\frac{1}{K}\sum_{k=1}^{K}\mathbb{1}\left(\widehat{\mathsf{QV}}_k(h) \geq \lambda - \sqrt{\frac{\log\frac{(K+1)}{\delta}}{2n_k}}\right) = \lim_{K\to\infty}\frac{1}{K}\sum_{k=1}^{K}\mathbb{1}\left(\lim_{n_k\to\infty}\widehat{\mathsf{QV}}_k(h) + \sqrt{\frac{\log\frac{(K+1)}{\delta}}{2n_k}} \geq \lambda\right)
$$
$$
= \lim_{K\to\infty}\frac{1}{K}\sum_{k=1}^{K}\mathbb{1}\left(\lim_{n_k\to\infty}\widehat{\mathsf{QV}}_k(h) + \lim_{n_k\to\infty}\sqrt{\frac{\log\frac{(K+1)}{\delta}}{2n_k}} \geq \lambda\right)
$$
$$
\overset{a.s.}{=} \lim_{K\to\infty}\frac{1}{K}\sum_{k=1}^{K}\mathbb{1}\left(\mathbb{E}_{P_k}\left[\ell\left(y, h\left(\boldsymbol{X}\right)\right)\right] \geq \lambda\right)
$$
$$
\overset{a.s.}{=} \mathbb{E}_{P\sim\mu}\left[\mathbb{1}\left(\mathbb{E}_P\left[\ell\left(y, h\left(\boldsymbol{X}\right)\right)\right] \geq \lambda\right)\right]
$$
$$
= \mu\left(\mathbb{E}_P\left[\ell\left(y, h\left(\boldsymbol{X}\right)\right)\right] \geq \lambda\right). \tag{100}
$$

Again, the *almost sure* equalities are the result of applying law of large numbers in both distributional and meta-distributional levels. Therefore, the proof for this part is complete.

**AMO for Theorems 5.2 and 5.3** : Assume two arbitrary meta-distributions $\mu'$ and $\mu$ and only assume $\mathcal{D}_f(\mu'\|\mu) \leq \varepsilon$ (note that for $\varepsilon < \infty$, this assumption automatically enforces relative continuity as well), and let $P_1, P_2, \ldots$ be an infinite i.i.d. sequence from $\mu$. The only extra restriction over $\mu'$ is that it represents a *probability measure*. Combining all these natural restrictions we get

- Non-negativity of $\mu'$, which means for any $\mathcal{S} \subseteq \mathcal{M}(\mathcal{Z})$ we have $\mu'(\mathcal{S}) \geq 0$.

- Normalization condition of probability measures which mathematically translates into the following equalities:

$$
1 = \mu'\left(\mathcal{M}(\mathcal{Z})\right) = \int_{P\in\mathcal{M}}\mu'(\mathrm{d}P) = \mathbb{E}_{P\sim\mu}\left(\frac{\mu'(\mathrm{d}P)}{\mu(\mathrm{d}P)}\right) \overset{a.s.}{=} \lim_{K\to\infty}\frac{1}{K}\sum_{k=1}^{K}\frac{\mathrm{d}\mu'}{\mathrm{d}\mu}(P_k).
$$

- Boundedness of $f$-divergence, which can be written as follows:

$$
\varepsilon \geq \mathcal{D}_f(\mu'\|\mu) = \mathbb{E}_{P\sim\mu}\left[f\left(\frac{\mathrm{d}\mu'}{\mathrm{d}\mu}(P)\right)\right] \overset{a.s.}{=} \lim_{K\to\infty}\frac{1}{K}\sum_{k=1}^{K}f\left(\frac{\mathrm{d}\mu'}{\mathrm{d}\mu}(P_k)\right).
$$

In all the above statements, the *almost sure* equality holds with respect to the randomness of drawing the i.i.d. sequence of distributions $P_1, P_2, \ldots$ from $\mu$ and utilizes the application of the law of large numbers. Based on the above discussions, any measure defined on a Borel $\sigma$-field over $\mathcal{M}(\mathcal{Z})$ that satisfies the above conditions is a potential candidate for the worst-case $\mu'$ in the statement of theorem.

On the other hand, the proposed bound in the right-hand side of the main result in Theorem 5.2 is computed via the following optimization problem:

$$\widehat{B}^*(\varepsilon) \triangleq \sup_{0 \leq \alpha_1, \ldots, \alpha_K \leq \Lambda} \frac{1}{K} \sum_{k=1}^{K} \alpha_k \widehat{\mathsf{QV}}_k(h) \tag{101}$$

$$\text{subject to} \quad \left| \frac{1}{K} \sum_{k=1}^{K} \alpha_k - 1 \right| \leq c_1 \sqrt{\log(\delta^{-1})/K},$$

$$\frac{1}{K} \sum_{k=1}^{K} f(\alpha_k) \leq \varepsilon + c_2 \sqrt{\log(\delta^{-1})/K}.$$

Since, for all $\delta > 0$ we have

$$\lim_{K \to \infty} c_1 \sqrt{\log(\delta^{-1})/K} = 0, \quad \text{and} \quad \lim_{K \to \infty} \varepsilon + c_2 \sqrt{\log(\delta^{-1})/K} = \varepsilon, \tag{102}$$

in the asymptotic regime of $K \to \infty$ the vector $(\alpha_1, \ldots, \alpha_K)$ converges to the sequence $\{\alpha_i\}_{i \in \mathbb{N}}$, with the following properties:

$$\alpha_i \geq 0, \ \forall i \in \mathbb{N}, \quad \lim_{K \to \infty} \frac{1}{K} \sum_{i=1}^{K} \alpha_i = 1 \quad \text{and} \quad \lim_{K \to \infty} \frac{1}{K} \sum_{i=1}^{K} f(\alpha_i) \leq \varepsilon. \tag{103}$$

Therefore, it can be deduced that any $\{\alpha_i\}_{i \in \mathbb{N}}$ that satisfies such conditions corresponds to an achievable density ratio, i.e. $\alpha_k = \mathrm{d}\mu'/\mathrm{d}\mu(P_k)$, $k \in \mathbb{N}$, for some $\mu' \in \mathcal{M}^2(\mathcal{Z})$. In other words, the sup value is achievable. The same set of arguments hold for Theorem 5.3, and the proof for this part is complete.

**AMO for Theorem 6.3** : The proof relies heavily on the theory developed in the proof of Theorem 6.3 (see Section E), and we adopt the same procedure and notations. Recall the bound proposed in Theorem 6.3: For any $\varepsilon, \delta > 0$, consider the following constrained optimization problem:

$$\widehat{U}^*(\varepsilon) \triangleq \sup_{\rho_1, \ldots, \rho_K \geq \varepsilon/K} \frac{1}{K} \sum_{k=1}^{K} \widehat{\mathsf{QV}}_k(h, \rho_k) \tag{104}$$

$$\text{subject to} \quad \frac{1}{K} \sum_{k=1}^{K} \rho_k \leq \varepsilon \left( 1 + \frac{1}{K} \right) + c_1 \sqrt{\frac{\log \left( \frac{K+2}{\delta} \right)}{K}}.$$

In the asymptotic regime (i.e., $K, \min_k n_k \to \infty$), we have

$$\lim_{n_k \to \infty} \widehat{\mathsf{QV}}_k(h, \rho_k) \overset{a.s.}{=} \sup_{Q \in \mathcal{B}_{\rho_k}(P_k)} \mathbb{E}_Q[\ell(y, h(\boldsymbol{X}))], \tag{105}$$

where $P_1, P_2, \ldots$ is an infinite (but countable) i.i.d. sequence from the meta-distribution $\mu$. The almost sure convergence follows from the law of large numbers. Define $\rho : \mathcal{M}_{\mathcal{Z}} \to \mathbb{R}_+$. In the asymptotic regime ($K \to \infty$), the constraint becomes:

$$\mathbb{E}_{P \sim \mu}[\rho(P)] \overset{a.s.}{=} \lim_{K \to \infty} \frac{1}{K} \sum_{k=1}^{K} \rho_k$$

$$\leq \lim_{K \to \infty} \varepsilon \left( 1 + \frac{1}{K} \right) + c_1 \sqrt{\frac{\log \left( \frac{K+2}{\delta} \right)}{K}}$$

$$= \varepsilon. \tag{106}$$

Thus, the optimization problem defining $\widehat{U}^*$ when $K, n_k \to \infty$ (almost surely) reduces to:

$$\widehat{U}^*(\varepsilon) \overset{a.s.}{=} \sup_{\rho : \mathcal{M}(\mathcal{Z}) \to \mathbb{R}_+} \mathbb{E}_{P \sim \mu} \left[ \sup_{Q \in \mathcal{B}_{\rho_k}(P_k)} \mathbb{E}_Q[\ell(y, h(\boldsymbol{X}))] \right]$$

$$\text{subject to} \quad \mathbb{E}_{P \sim \mu}[\rho(P)] \leq \varepsilon. \tag{107}$$

Introducing a Lagrange multiplier $\gamma \geq 0$, and using strong duality, this problem becomes:

$$\inf_{\gamma \geq 0} \left\{ \sup_{\rho:\mathcal{M}(\mathcal{Z}) \to \mathbb{R}_+} \mathbb{E}_{P \sim \mu} \left[ \sup_{Q \in \mathcal{B}_{\rho(P)}(P)} \mathbb{E}_Q \left[ \ell\left(y, h\left(\boldsymbol{X}\right)\right) \right] \right] - \gamma \left( \mathbb{E}_{P \sim \mu} \left[ \rho\left(P\right) \right] - \varepsilon \right) \right\}$$

$$= \inf_{\gamma \geq 0} \left\{ \sup_{\rho:\mathcal{M}(\mathcal{Z}) \to \mathbb{R}_+} \mathbb{E}_{P \sim \mu} \left[ \sup_{Q \in \mathcal{B}_{\rho(P)}(P)} \mathbb{E}_Q \left[ \ell\left(y, h\left(\boldsymbol{X}\right)\right) \right] - \gamma \left( \rho\left(P\right) - \varepsilon \right) \right] \right\}$$

$$= \inf_{\gamma \geq 0} \left\{ \gamma\varepsilon + \sup_{\rho:\mathcal{M}(\mathcal{Z}) \to \mathbb{R}_+} \mathbb{E}_{P \sim \mu} \left[ \sup_{Q \in \mathcal{B}_{\rho(P)}(P)} \mathbb{E}_Q \left[ \ell\left(y, h\left(\boldsymbol{X}\right)\right) \right] - \gamma\rho\left(P\right) \right] \right\}. \tag{108}$$

The combined effect of the two supremum operators ($\sup_{\rho:\mathcal{M}_\mathcal{Z} \to \mathbb{R}_+}$ and $\sup_{Q \in \mathcal{B}_{\rho(P)}(P)}$) yields the following min-max problem:

$$\inf_{\gamma \geq 0} \left\{ \gamma\varepsilon + \mathbb{E}_{P \sim \mu} \left[ \sup_{Q \in \mathcal{M}_\mathcal{Z}} \mathbb{E}_Q[\ell\left(y, h\left(\boldsymbol{X}\right)\right)] - \gamma \mathcal{W}_c(P, Q) \right] \right\}. \tag{109}$$

By Lemma E.1, this is equal to:

$$\sup_{\mu' \in \mathcal{G}_\varepsilon(\mu)} \mathbb{E}_{P \sim \mu'} \left( \mathbb{E}_P[\ell\left(y, h\left(\boldsymbol{X}\right)\right)] \right). \tag{110}$$

Thus, the proposed bound $\widehat{U}^*$ converges to the above expression as $K, \min_k n_k \to \infty$, which is exactly equal to the statistical quantity that it was supposed to upper-bound. This completes the proof. $\qquad\square$

## G. Complementary Experimental Results

In this section, we present a complementary series of experiments on real-world datasets to demonstrate: (i) the validity and tightness of our bounds, and (ii) that our bounds are not only theoretically sound but also practical to compute. For clarity and readability, we repeat key explanations from Section 8 to maintain a cohesive narrative.

### G.1. Client Generation and Risk CDF Certificates for Unseen Clients

In the first part of our experiments, we outline our client generation model and present a number of risk CDF guarantees. We simulated a federated learning scenario with $n = 1000$ nodes, where each node contains 1000 local samples. The experiments were conducted using four different datasets: CIFAR-10 (Krizhevsky et al., 2009), SVHN (Netzer et al., 2011), EMNIST (Cohen et al., 2017), and ImageNet (Russakovsky et al., 2015). To create each user's data within the network, we applied three types of affine distribution shifts across users:

- **Feature Distribution Shift:** Each sample $\boldsymbol{X}_i^{(k)}$ in the dataset is manipulated via a transformation chosen randomly for each node. Specifically, each user is assigned a unique matrix $\Lambda^{(k)}$ and shift vector $\boldsymbol{\delta}^{(k)}$, and the data is modified as follows:
$$\widetilde{\boldsymbol{X}}_i^{(k)} = (I + \Lambda^{(k)})\boldsymbol{X}_i^{(k)} + \boldsymbol{\delta}^{(k)}. \tag{111}$$
In our experiments, $\Lambda^{(k)}$ and $\boldsymbol{\delta}^{(k)}$ are respectively random matrices and vectors with i.i.d. zero-mean Gaussian entries. The standard deviation varies based on the dataset: 0.05 for CIFAR-10 and SVHN, 0.1 for EMNIST, and 0.01 for ImageNet.

- **Label Distribution Shift:** Here, we assume that each meta-distribution is characterized by a specific $\alpha$ coefficient. To generate each user's data under this shift, the number of samples per class is determined by a Dirichlet distribution with parameter $\alpha$. In our experiments, we use $\alpha = 0.4$.

- **Feature & Label Distribution Shift:** As the name suggests, this shift combines both the feature and label distribution shifts described above to create a new distribution for each user.

Figure 4 illustrates our performance certificates (i.e., bounds on the risk CDF) for unseen clients when there are no shifts. We selected 100 nodes from the population and considered 400 other nodes as unseen clients. We then plotted the CDFs based on 100 samples and confirmed that our bounds hold for the real population as well. Due to the standard DKW inequality, the empirical CDF is a good estimate for the test-time non-robust risk CDF.

## G.2. Certificates for $f$-Divergence Meta-Distributional Shifts

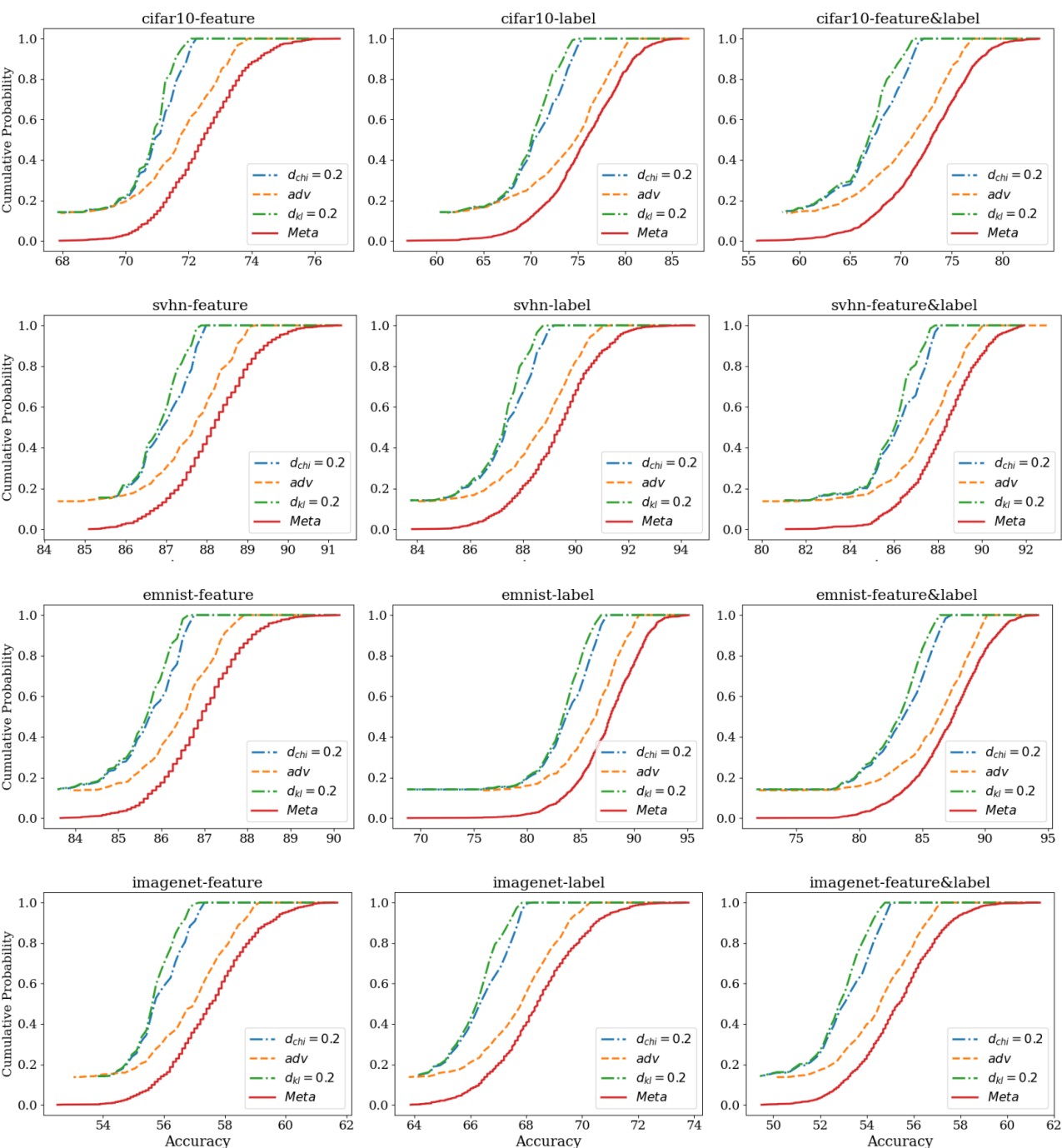

*Figure 9.* Extension of previous experiments to a broader range of adversarial budgets for $f$-divergence attacks. DKW-based certificates for unseen clients in our four examined datasets. *Meta* refers to the main population with 1000 nodes.

In this section, we examined scenarios where users belong to two distinct meta-distributions: the source and the target. A DNN-based model is initially trained on a network of clients sampled from the source. The resulting risk values are then fed into the optimization problems introduced in Section 5 to obtain robust CDF bounds, considering both the Chi-Square and KL divergence as potential choices for $f$. Finally, we empirically estimate the risk CDF for users from the target

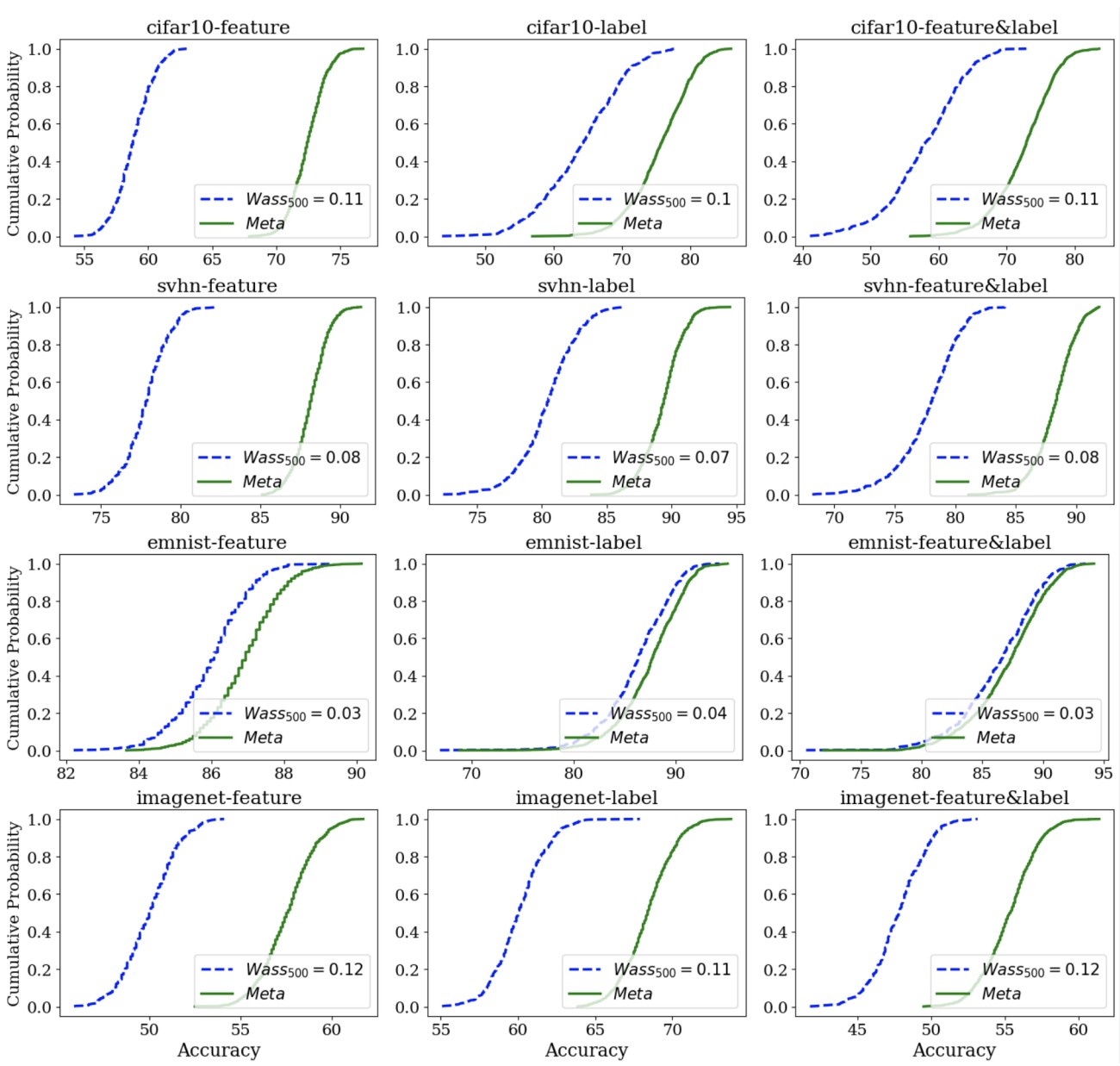

*Figure 10.* Wasserstein distance-based certificates for unseen clients in our four examined datasets. *Meta* refers to the main population with 1000 nodes. Dotted curves are based on 500 networks within the population.

meta-distribution and validate our bounds. Specifically, we tested our certificates in two distinct settings using the CIFAR-10 dataset (see Figure 5). We generated various image categories with differing resolutions or color schemes, and then sampled from these categories to create different distributions:

- **Resolutions**: Images were cropped and resized to create eight different resolutions. The Dirichlet $\alpha$ coefficients for the first (source) meta-distribution range from $0.4$ to $0.7$ for the four lower resolutions and from $0.7$ to $1$ for the four higher resolutions. For the second (target) meta-distribution, the ranges are reversed: $0.7$ to $1$ for the lower resolutions and $0.4$ to $0.7$ for the higher resolutions. The number of samples per resolution for each user is determined using a Dirichlet distribution, with $\alpha$ coefficients randomly selected from the specified range for the meta-distribution. As a result, users sampled from source will have more high-resolution images, while users from the target will have more low-resolution

samples.

- **Colors**: The color intensity of the images varies from $0.00$ (gray-scale) to $1.00$ (fully colored). For the source meta-distribution, the $\alpha$ coefficients range from $0$ to $0.5$ for images with color intensity below $0.5$, and from $0.5$ to $1$ for images above $0.5$. As with the resolution setting, the ranges are reversed for the target, and the number of samples per color intensity for each user is calculated similarly. Therefore, users sampled from source will have more colorful images, while those from the target will have more gray-scale images.

Figure 6 (left) verifies our CDF certificates based on both chi-square and KL-divergence (dotted curves) for the target meta distribution (orange curve). As can be seen, bounds have tightly captured the behavior of risk CDF in the target network. More detailed experiments are shown in Figure 9 in Appendix G.

### G.3. Certificates for Wasserstein-based Meta-Distributional Shifts

In this experiment, as previously mentioned, we used affine distribution shifts to create new domains. Figure 6 (right) summarizes our numerical results in this scenario. To generate different networks within the meta-distribution, we applied the affine distribution shifts described in Section G.1. Once again, the results validate our certificates, this time for Wasserstein-type shifts. The blue curve, representing the real population, consistently falls within or beneath the blue shaded area. Regarding tightness, it is important to note that the bounds presented here remain tight, particularly under adversarial attacks as defined by a distributional adversary in (Sinha et al., 2018). More detailed experiments with various levels of tightness are shown in Figure 10 in Appendix G.

Although our theoretical findings in Section 6 focus solely on the average risk and not the risk CDF, we extended the same framework to the CDF in this experiment to explore whether the theory might also apply. The results were positive, suggesting potential for extending our theoretical findings in this area. A more detailed series of simulation results are presented via Figures 9 and 10, respectively.

Figure 9 illustrates complementary results for $f$-divergence bounds on risk CDF, which are robust extensions of the DKW bound. Simulations have been repeated for several different values of $\varepsilon$, where both KL and $\chi^2$ (Chi-Square) divergences have been considered.

Figure 10 shows the robust CDF bounds for Wasserstein shifts. Again, different values of $\varepsilon$ for Wasserstein distance have been considered which have resulted in several bounds with various levels of tightness. It should be noted that all bounds are asymptotically minimax optimal, meaning that an adversary can choose a distribution to perform exactly as bad as the bounds, at least in the asymptotic regime where both $K$ and $\min_k n_k$ tend to infinity.

