# OpenReview forum: "Certifiably Robust Model Evaluation in Federated Learning under Meta-Distributional Shifts"
_ICML.cc/2025/Conference — ICML 2025 poster_

### Official Review · Reviewer_SPhP · 2025-03-10

**Overall Recommendation:** 4

**Summary:**

The paper tackles the problem of robust federated evaluation, in which a server aims to evaluate a model $h$ on the private federated data while taking into account the scenario in which the model could be used on a different distribution in deployment. The paper provides theoretical bounds to characterize the federated empirical evaluation of the model while using only a reasonable number of queries from the server to the clients. Two main Meta-Distributional Shifts were analyzed: f-divergence and Wasserstein distance shifts. Experimental results were provided to show the tightness of the provided bounds.

**Claims And Evidence:**

Most of the claims are backed with clear evidence, with the exception of the following :

- In Remark 7.3, the authors claim as a certificate of privacy the absence of current attacks capable of recovering private data from the loss queries. This is not a certificate, contrary to information theoretical privacy certifications such as differential privacy.  For instance, for the computation in Equation 15, given that the number of queries from the server to the clients can be large, it is more than likely that the privacy of local data can be compromised.

**Essential References Not Discussed:**

None that I know of.

**Experimental Designs Or Analyses:**

The authors provide experiments covering non-robust as well as the two Meta-Distributional Shifts. Settings not covered by the theory are also included. However, it is not clear what the authors mean by tightness in the interpretation of the results. Indeed, Figure 6, for instance, shows quite a large difference between the CDFs.

**Methods And Evaluation Criteria:**

The server solves constrained convex (or quasi-convex) optimization problems to compute the desired empirical quantities. For Wasserstein Meta-distributional shifts, the client is also required to solve a convex optimization problem. However, the paper does not clarify how an imprecise solution to this optimization problem, with error $\Delta$, can be used on the server side. In other words, it is assumed that the clients can compute $\widehat{QV_k}(h, \rho_k)$ precisely, which is not always possible.

**Other Comments Or Suggestions:**

Small typo:
- 297 (left) two meta-distributions µ, µ’

The paper should also include an impact statement, as required by the ICML 2025 guidelines.

**Other Strengths And Weaknesses:**

I included the key strengths and weaknesses in the previous paragraphs.

**Questions For Authors:**

How would you explain the gap between the CDFs in Figure 6 ?

**Relation To Broader Scientific Literature:**

The contributions of this paper seem novel and relevant to the emerging field of federated evaluation.

**Theoretical Claims:**

I did not check the correctness of the proofs.

---

> ### Author Rebuttal · Authors · 2025-03-31
>
> We sincerely thank the reviewer for their careful analysis and insightful comments on our paper. Below, we provide point-by-point responses to each concern.
>
> --------------------------
>
> **Claims and Evidence**
>
> - **"Remark 7.3 is not an actual certificate."**
>   The reviewer is absolutely correct. We will remove the term "certificate" from Remark 7.3 and adopt a more precise phrasing. Our work does not focus on privacy analysis from a differential privacy (DP) perspective. Instead, Remark 7.3 simply states that, to the best of our knowledge, no known model inversion attack can reconstruct samples solely from a series of adversarial loss values computed at different radii.
>
> ----------
>
> **Methods and Evaluation Criteria**
>
> - **"The paper does not clarify how an imprecise solution to this optimization problem, with error $\Delta$, can be used on the server side. In other words, it is assumed that the clients can compute $\widehat{\mathsf{QV}}(h,\rho)$ precisely, which is not always possible."**
>   Thank you for pointing this out. The reviewer is correct that an imprecise computation of $\widehat{\mathsf{QV}}(h,\rho)$ introduces an additional error term. However, since the empirical bound in Theorem 6.3 is based on an average of QVs, an approximation error of at most $\Delta$ per QV results in a total additional error of at most $\Delta$. Thus, the generalization gap should include two components:
>   i) The **statistical residual**, already formulated in Theorem 6.3.
>   ii) An **algorithmic residual**, which can be arbitrarily reduced by (polynomially) increasing the computational effort.
>   We will explicitly incorporate this detail into the revised version.
>
> ------------------
>
> **Experimental Design and Analyses**  and also **Questions**:
>
> - **"How would you explain the gap between the CDFs in Figure 6?"**
>
>   Thank you for bringing this to our attention. As the reviewer correctly noted, the gap depends on how "tightness" is interpreted. The current gap shown in the paper should not vanish because the difference between the CDFs of evaluation and target clients remains nonzero for any strictly positive $\varepsilon$. However, by "tightness," we specifically refer to the **generalization gap** approaching zero.
>
>   In Figure 6, we currently show our bounds alongside the empirical CDF of loss over evaluation clients (without adversarial attacks) and compare them to one specific attack (modifying image coloring and resolution). However, our bounds hold for **the worst-case** scenario. To better illustrate this, we have added an extra curve representing **an achievable, but more general** adversarial performance. This curve aligns more closely with the theoretical bounds, making the tightness clearer.
>
>   You can view the updated plots here:
>   [GitHub Link: https://github.com/annonymous-ICML2025/paper-3310](https://github.com/annonymous-ICML2025/paper-3310)
>
>   This will show that our bounds closely match these worst-case adversarial shifts, with any remaining gap diminishing as $K$ and $\min_k n_k$ increase. We will update the plots accordingly in the revised version.
>
> -------------------
>
> We welcome further discussion and, once again, appreciate your time and effort.

---

### Official Review · Reviewer_K26v · 2025-03-11

**Overall Recommendation:** 3

**Summary:**

The paper proposes algorithms to robustly estimate a model performance under a ball of f-divergence or Wasserstain distance, and provide the generalization bounds of the proposed methods.

**Claims And Evidence:**

The paper claims the proposed methods can estimate model performance robustly, though it is mainly a theory work, the claim is supported by some experiments.

**Essential References Not Discussed:**

N/A

**Experimental Designs Or Analyses:**

The experiments are using simulated data with datasets CIFAR-10, SVHN, EMNIST, and ImageNet. For each dataset a client-dependent transformation is applied to create heterogeneous data distribution. Then proposed methods are used to estimate CDF for model evaluation and it is compared with empirical CDF. The proposed methods can capture shape and trends of the true CDF.

**Methods And Evaluation Criteria:**

Yes.

**Other Comments Or Suggestions:**

N/A

**Other Strengths And Weaknesses:**

N/A

**Questions For Authors:**

N/A

**Relation To Broader Scientific Literature:**

It tackles the problem of robustly estimating model performance in a federated environment where some clients may be out of the network during evaluation. It is a relevant setting in federated learning.

**Theoretical Claims:**

The main claims are the derived risk and CDF of the proposed estimators, showing their convergence rate to the true value as the number of clients and samples grows. I did not check proofs in the appendix.

---

> ### Author Rebuttal · Authors · 2025-03-31
>
> We sincerely thank the reviewer for their insightful and positive comments on our paper. Please let us know if you have any further questions or concerns—we would be happy to address them and clarify any points that might help you further increase your score.

---

### Official Review · Reviewer_6ciV · 2025-03-12

**Overall Recommendation:** 2

**Summary:**

This paper introduces a robust optimization framework for evaluating machine learning models in federated settings with non-IID client data, where the data distributions are governed by an unknown meta-distribution. The goal is to assess model performance not only on a given client network (standard A/B testing) but also on unseen networks with similar distributions, measured using f-divergence or Wasserstein distance. The framework enables private server-side aggregation of local adversarial risks, ensuring robust global model evaluation with polynomial time and query complexity. Theoretical results establish minimax-optimal risk bounds with vanishing generalization gaps as the network size increases. Empirical evaluations confirm the framework’s effectiveness in real-world tasks.

**Claims And Evidence:**

The motivation and intuition of the proposed methods are not adequately discussed.

**Essential References Not Discussed:**

No.

**Experimental Designs Or Analyses:**

Yes.

**Methods And Evaluation Criteria:**

Yes.

**Other Comments Or Suggestions:**

See above.

**Other Strengths And Weaknesses:**

Strength:

Analyzed the bound of generalization in evaluating models in federated learning. Different distribution shifts have been considered such as f-divergence and Wasserstein distributional shift. An optimization framework was developed to get a generalization bound for model evaluation in federated learning.

Weakness:

1. The theorems analyze bounds under different distributions. Can the authors please highlight the key challenges of these analysis in federated learning compared to non-federated learning?

2. It seems that it lacks an analysis of complexity to solve the proposed optimization problem, for both the global server and local clients.

3. The writing could be improved by more discussion of the motivation and intuition of the proposed optimization framework.

4. The experiments can hardly demonstrate that the bounds are tight. It is questionable whether it could provide an effective evaluation in reality.

**Questions For Authors:**

See above.

**Relation To Broader Scientific Literature:**

Related to federated leanring.

**Theoretical Claims:**

No, I did not.

---

> ### Author Rebuttal · Authors · 2025-03-31
>
> We sincerely thank the reviewer for their insightful comments. Below, we provide point-by-point responses to each concern.
>
> -----------------------------
>
> **Questions**
>
> 1) **"The theorems analyze bounds under different distributions. Can the authors please highlight the key challenges of this analysis in federated learning compared to non-federated learning?"**
> Our analysis evaluates model performance under different (meta-)distributional shifts. The key challenge in federated learning (FL) is that data is decentralized and private, unlike centralized settings where full data access is available. Our bounds are computable solely using Query Value (QV) functions (defined in Section 3), meaning we do not require access to clients' private data or even their local sample sizes. This makes our method particularly suited for federated model evaluation. In contrast, existing methods either compromise privacy (by accessing raw data) or suffer from non-vanishing assessment gaps. For further details, please refer to our response to Reviewer nqsu.
>
> 2) **"It seems that an analysis of complexity to solve the proposed optimization problem, for both the global server and local clients, is missing."**
> In most parts of Section 7, and almost all of Appendix F (proofs), we have provided a thorough **computational complexity analysis** for both server-side and client-side optimizations. In particular, Remark 7.1 discusses server-side complexity, while Remark 7.2 covers client-side computational aspects. Below, we summarize the key results:
>
> - **Server-side:**
>   - The non-robust case in Theorem 4.1 only requires summing queried loss values, without any optimization.
>   - The optimization problems in Theorems 5.2 and 5.3 are convex, featuring linear objectives with either linear or linearly separable convex constraints. Such problems are solvable in polynomial time with at least a linear convergence rate (see the proof of Remark 7.1 in Appendix F).
>   - The optimization problem in Theorem 6.3 is quasi-convex. It is solved using the bisection algorithm in Algorithm 1 (Appendix F), which finds the global optimum in logarithmic time. Each step of this algorithm involves solving a convex program with at least a linear convergence rate (again, see the proof of Remark 7.1 in Appendix F).
>
> - **Client-side:**
>   - Theorems 4.1, 5.2, and 5.3 require no client-side optimization; clients only compute their local non-robust loss.
>   - Theorem 6.3 involves a local optimization at the client side to assess adversarial loss. This problem has been well studied (see proof in Remark 7.2, based on Sinha et al. (2017)), and under the given assumptions, it is convex with at least a linear convergence rate.
>
> 3) "**The writing could be improved by more discussion of the motivation and intuition of the proposed optimization framework.**": This issue has already been addressed in our response to Reviewer nqsu. Please refer to our response to Question 4 (illustrative example).  We make sure to highlight such examples in the revised version.
>
> **4) "**The experiments can hardly demonstrate that the bounds are tight.**":** Thank you for pointing this out. The bounds are indeed tight, but the figures required an additional detail to illustrate this more clearly. We have addressed this in our response to Reviewer SPhP, who raised a similar concern. Please refer to that discussion for further details. To clarify, the figures in the paper compare the source non-adversarial setting with one specific adversary (modifying image coloring and resolution), while the bounds hold for the **worst-case** scenario. To better demonstrate the claimed tightness, we have added an extra curve to the plots, representing **an achievable** adversarial example based on the source network. This addition makes the tightness more apparent. You can view the updated figures here:
> [GitHub Link: https://github.com/annonymous-ICML2025/paper-3310](https://github.com/annonymous-ICML2025/paper-3310)
> We will also incorporate the necessary graphical details in the revised version of the paper.
>
> ------------------------
>
> We welcome further discussion and, once again, appreciate your time and effort. If our responses have been satisfactory, we would greatly appreciate it if you reconsider your score.

---

### Official Review · Reviewer_nqsu · 2025-03-16

**Overall Recommendation:** 3

**Summary:**

The paper provides an analysis of model evaluation under tighter risk assessment conditions compared to previous works on federated evaluation. The main contribution includes a novel extension of the Dvoretzky–Kiefer–Wolfowitz (DKW) inequality adapted for federated data distributions. The authors claim improved evaluation procedures leveraging the federated nature of data, emphasizing the impact of meta-distribution similarity on evaluation quality.

**Claims And Evidence:**

Most claims are supported by rigorous theoretical proofs and experimental validation. However, some claims about the inherent wellness or robustness of risk analysis compared to prior work need clearer justification. In particular, the paper does not adequately explain why previous methods ignored inherent robustness or whether their analysis was suboptimal due to different assumptions or scenarios.

**Essential References Not Discussed:**

I am not aware of any missing references, although, I am not very familiar with all the relevant literature.

**Experimental Designs Or Analyses:**

See Methods And Evaluation Criteria.

**Methods And Evaluation Criteria:**

The evaluation makes sense in the context that the main contribution of this work is theoretical.

**Other Comments Or Suggestions:**

1. Improve the clarity by explicitly defining all notations and assumptions in a dedicated section.

2. Enhance discussions around practical implications, especially regarding meta-distribution similarity handling.

**Other Strengths And Weaknesses:**

The primary strength of the paper lies in its theoretical innovation and careful extension of known statistical bounds (DKW inequality) into federated learning contexts. However, the clarity of presentation suffers at points due to missing explicit definitions of notations and assumptions, making parts of the paper challenging to follow.

**Questions For Authors:**

1. Could you clarify precisely why previous methods did not sufficiently address inherent wellness or robustness of risk? Are there scenarios they overlooked, or did their approaches inherently limit their analysis?

2. Can you explicitly demonstrate or discuss how your results benefit specifically from the federated nature of the data compared to centralized settings? In particular, does evaluation benefit from having more clients, although with limited data?

3. Could you elaborate more on the practical implications and handling of meta-distribution similarity in real-world scenarios? How would practitioners estimate or manage this similarity effectively?

**Relation To Broader Scientific Literature:**

The paper situates itself well within the broader context of federated learning and risk analysis, extending prior results with more refined statistical bounds. However, the authors could strengthen this section by explicitly contrasting their contributions against key previous works on robustness and risk evaluation in federated learning.

**Theoretical Claims:**

I have reviewed the theoretical claims, and they appear to be correct.

---

> ### Author Rebuttal · Authors · 2025-03-31
>
> We sincerely thank the reviewer for their insightful comments.  Here, we give point-to-point responses to each concern or question.
>
> ---------------------------------
>
> **Main Comments**
>
> The reviewer’s main concerns are:
> i) Justifications for our claim that previous methods either ignore the inherent robustness or wellness of the evaluated model or are suboptimal due to restrictive assumptions.
> ii) Missing explicit definitions of notations and assumptions.
>
> **Regarding (i):**
> We have provided a detailed explanation in Section A of the Appendix due to page limitations. However, as requested, we can move it to the main body. In Section A, we categorize existing methods into two approaches:
> a) They collect all client samples on a central server, approximate the meta-distribution (e.g., via histograms), and design collective (meta-)distributional attacks to assess adversarial loss. This approach obviously violates privacy in federated learning (FL) and is impractical for real-world model evaluation, making it primarily useful only for research purposes (See Reisizadeh et al., 2020; Ma et al., 2024 for more details).
> b) They introduce a "model-independent" additive term, often using Maximum Mean Discrepancy (MMD) (e.g., in Sinha et al (2017), Pillutla et al. (2024), etc.), to inflate the loss. This approach is suboptimal because it disregards the model's inherent robustness. Consequently, the generalization gap does not vanish as the sample size increases, meaning a constant, non-vanishing term must always be added to the evaluation network's loss to certify robustness for the target network.
>
> **Regarding (ii):**
> We have included as many definitions as space allows, but we can reorganize them into a dedicated section for clarity, as suggested by the reviewer. Regarding assumptions, we make none beyond assuming that local distributions are independent samples from a meta-distribution. All other aspects are kept as general as possible. If the reviewer has specific assumptions in mind, please let us know.
>
> ------------------------------------
>
> **Questions**
>
> 1) Addressed above.
>
> 2) **"How do your results specifically benefit from the federated nature of the data compared to centralized settings?"**
> Our bounds are computable solely using Query Value (QV) functions (defined in Section 3), meaning we do not require access to clients' private data or even their local sample sizes. In contrast, existing methods either compromise privacy (by accessing raw data) or suffer from non-vanishing assessment gaps.
>
> 2) (2, cont'd) **"Does evaluation benefit from having more clients, even if each has limited data?"**
> Yes! A larger number of clients $K$ provides more information about the underlying meta-distribution $\mu$, even when their data remains private. Consequently, even if each client has limited data, (at least) parts of our bounds asymptotically improve as $K$ increases. This has been theoretically shown in our results in Theorems 4.1, 5.2, 5.3 and 6.3. We will emphasize this in the revised version.
>
> 3) **Illustrative Example:**
> Consider a company planning to launch an app for a target customer base in New York. Before full deployment, they might conduct a pilot test in a smaller, controlled community (e.g., New Jersey) to evaluate user experience, infrastructure capabilities, etc. A key challenge is ensuring that insights from New Jersey generalize to New York, given possible differences in user lifestyles and preferences between the two cities. Our work addresses this by providing privately and efficiently computable bounds that extend beyond the evaluation network to unseen (but similar) target networks. How to choose $\varepsilon$ in practice? This highly depends on the application. However, the main point is usually to see how fast the performance of $h$ "degrades" as $\varepsilon$ grows, which is a sign of the sensitivity of the model. This does not need the exact knowledge of $\varepsilon$ in practice.
>
> ----------------------------
>
> We welcome further discussion and, once again, appreciate your time and effort. If our responses have been satisfactory, we would greatly appreciate it if you increase your score.

---

> > ### Comment · Reviewer_nqsu · 2025-04-09
> >
> > Thank you for addressing some of my concerns, I increased my score from 2 to 3.
> >
> > I would further request that the authors clarify:
> >
> > 1. Your evaluation does not necessarily benefit from increasing client size $K.$ As you answered in the rebuttal, only part of the bound improves with increasing K, but the leading term might actually worsen, see Thm 4.1 and discussion below. Is this expected and tight, or is this just the limit of your current analysis?
> >
> > 2. You say that the prior literature performs an evaluation by approximating the meta-distribution (e.g., via histograms). As far as I know, this could be approximated using federated analytics [1] that uses differential privacy and does require direct access to clients' data.
> >
> > [1] Xu, Zheng, et al. "Federated Learning and Analytics in Practice: Algorithms, Systems, Applications, and Opportunities." International Conference on Machine Learning. 2023.

---

### Decision · Program_Chairs · 2025-05-01

**Decision:**

Accept (poster)

**Comment:**

The paper proposes a robust optimization framework for evaluating machine learning models in federated settings where client data is non-IID and drawn from an unknown meta-distribution. Unlike standard evaluation approaches that consider only the current client network, the proposed method aims to generalize model evaluation to unseen but distributionally similar client networks. This is achieved through private server-side aggregation of local adversarial risks, incorporating f-divergence and Wasserstein distance measures for robustness. Theoretical contributions include asymptotically minimax-optimal bounds for risk and CDF estimation, with provable vanishing generalization gaps as the number of clients and sample sizes increase. The framework is designed to be computationally efficient and privacy-conscious, with empirical results showing its effectiveness across multiple real-world datasets.

Reviewers generally acknowledged the paper’s strong theoretical foundation, with particular appreciation for its novel application of the DKW inequality in the federated learning context and its careful treatment of meta-distributional shifts. Multiple reviewers highlighted the contribution’s relevance to federated evaluation and its potential to advance understanding of generalization under distributional shifts. However, some concerns were raised about the presentation and the lack of detailed intuition behind the proposed optimization framework. Several reviewers also noted missing complexity analysis especially given the number of server queries involved. Most of the reviewers' concerns are addressed by the authors' responses. However, some of the reviewers are still not convinced by the practicality of the derived bounds since the nature of the bounds are min-max and might be pessimistic in practice.